# Utilizing cost-effective pyrocarbon for highly efficient gold retrieval from e-waste leachate

Kaixing Fu[1], Xia Liu[2], Xiaolin Zhang [ID][3], Shiqing Zhou[4], Nanwen Zhu[1], Yong Pei[5] & Jinming Luo [ID][1] [✉]

Addressing burdens of electronic waste (E-waste) leachate while achieving sustainable and selective recovery of noble metals, such as gold, is highly demanded due to its limited supply and escalating prices. Here we demonstrate an environmentally-benign and practical approach for gold recovery from E-waste leachate using alginate-derived pyrocarbon sorbent. The sorbent demonstrates potent gold recovery performance compared to most previously reported advanced sorbents, showcasing high recovery capacity of 2829.7 mg g$^{-1}$, high efficiency (>99.5%), remarkable selectivity ($K_d$ ~ 3.1 × 10$^8$ mL g$^{-1}$), and robust anti-interference capabilities within environmentally relevant contexts. The aromatic structures of pyrocarbon serve as crucial electrons sources, enabling a hydroxylation process that simultaneously generates electrons and phenolic hydroxyls for the reduction of gold ions. Our investigations further uncover a "stepwise" nucleation mechanism, in which gold ions are reduced as intermediate gold-chlorine clusters, facilitating rapid reduction process by lowering energy barriers from 1.08 to −21.84 eV. Technoeconomic analysis demonstrates its economic viability with an input-output ratio as high as 1370%. Our protocol obviates the necessity for organic reagents whilst obtaining 23.96 karats gold product from real-world central processing units (CPUs) leachates. This work introduces a green sorption technique for gold recovery, emphasizing its role in promoting a circular economy and environmental sustainability.

The escalating use of electronic devices has resulted in E-waste emerging as the fastest-growing solid waste in the 21st century, boasting an annual growth rate of approximately 4%[1–3]. Aside from posing a significant threat to the environment and human health, E-waste is recognized as an urban mine due to its substantial economic potential, harboring non-renewable resources, such as gold (~280 g gold/ton), a content 10 to 100 times higher than that found in ores[2,4]. Significantly, the hydrometallurgical digestion solution produced from E-waste pretreatment process contains a comparatively high gold concentration (up to 2000 mg L$^{-1}$), which is considered an ideal source and carrier that entail efficient gold recovery. Recycling gold from such waste steams can concurrently mitigate environmental burden while yielding substantial economic gains, but their complexity requires a highly selective and sustainable recycling strategy. In comparison to conventional methods (e.g., solvent extraction[5] and electrochemical reduction[6]) that involve laborious procedures and expensive reagents, adsorption holds greater promise for scaled applications in recycling gold ions from water streams. Its advantages lie in low cost, ease of

[1]State Environmental Protection Key Laboratory of Environmental Health Impact Assessment of Emerging Contaminants, School of Environmental Science and Engineering, Shanghai Jiao Tong University, Shanghai 200240, China. [2]College of Chemistry and Chemical Engineering, Qingdao University, Qingdao 266071, China. [3]State Key Laboratory of Pollution Control and Resource Reuse, School of the Environment, Nanjing University, Nanjing 210023, China. [4]Hunan Engineering Research Center of Water Security Technology and Application, College of Civil Engineering, Hunan University, Changsha 410082, China. [5]Department of Chemistry, Xiangtan University, Xiangtan 411105, China. [✉]e-mail: jinming.luo@sjtu.edu.cn

operation, and the absence of no secondary pollution[7]. However, due to the complex hydrochemical conditions of E-waste leachate, traditional sorbents (e.g., silica[8], activated carbon[9] and resin[10,11]) encounter certain challenges in applications, including unsatisfactory gold selectivity against coexisting metal ions, potential structural collapse, and detachment of functional groups, and etc[12]. The development and synthesis of innovative sorbents with intricate microstructures and reactivity, such as porous polymer materials[13–16] and metal-based nanocrystals[17–19], have demonstrated enhanced gold adsorption from E-waste leachate. Nonetheless, these sorbents often face obstacles in large-scale production, characterized by low yield, expensive and intricate synthesis processes, and the generation of acid/organic wastewater[14,20]. These factors inevitably undermine the economic benefits associated with gold recycling, underscoring the pressing need for the development of a practical and sustainable sorbent with high economic viability.

Pyrocarbon, derived from pyrolyzed renewable biomass, owns various properties such as abundant sources, cost-effectiveness, and self-functionalization[21–24]. These characteristics make it particularly suitable for diverse environmental applications, with recent emphasis on its potential for gold recovery. Efforts have been made to extract gold ions using various pyrocarbon-based materials, such as activated carbon[25,26], carbon nanotubes[27,28], biochar[29,30], and graphene oxide[31–33], with predominant focus on the regulation of mesoporous structures or functional groups conducive to sorption (e.g., phenolic hydroxyl[32], sulfhydryl groups[34]). However, the limited understanding of the structure-function relationship of these pyrocarbon sorbents has constrained their efficacy and practical deployment in gold recovery operations. Generally, the sorption behaviors and underlying mechanisms (e.g., structure-function relationship, electron sources, gold reduction process) still remain blurred and elusive. This unclear relationship limited the application of pyrocarbon in practical gold recovery with the aspects of unsatisfactory performance, cost burden and odious impacts on environments. Notably, pyrocarbon itself is capable of reducing gold ions due to its inherent phenolic hydroxyl groups, which exhibits a profound capacity for electron donation through redox transformations (phenol⇌quinone)[35,36]. In addition, both the reductive activity and conductivity of graphitic domains in high-temperature pyrocarbon has been identified[21,37–39], suggesting their potential as electron donors and mediators that augment the supply of electrons for gold reduction. These properties are intimately associated with the intrinsic surface functionality, graphitization degree and conductivity characterized with pyrocarbon itself. In this context, optimizing and/or balancing these critical factors holds immense potential in significantly enhancing gold recovery using pyrocarbon, an area that offers important cost and environmental benefits but has received limited research attention to date.

This study introduces an eco-friendly alginate-derived pyrocarbon designed for highly efficient and selective gold recovery from E-waste leaching solution. By meticulously tailoring carbon hybridization levels and functional groups of pyrocarbon through the adjustments of pyrolysis temperatures, we achieved a notable capacity (2829.7 g g$^{-1}$) and swift kinetics ($k_f \sim 1.73 \times 10^{-8}$ m s$^{-1}$) for gold recovery. The extensive applicability of this pyrocarbon was validated through exceptional performance across various environmentally relevant conditions, encompassing gold concentrations, pH levels, ionic strengths, and real leaching solutions from CPUs. Furthermore, the synergistic application of operando spectroscopy in tandem with density functional theory (DFT) calculations unveiled both the electron sources and transfer pathways on the pyrocarbon. This approach also revealed a "stepwise" reduction mechanism for gold nucleus formation. By conducting a technoeconomic analysis (TEA), we demonstrated the cost benefits associated with the proposed pyrocarbon-based gold recovery process, showcasing an impressive input-output ratio of 1370%. Notably, this ratio could potentially exceed 2000% with

a 32% decrease in the price of CPUs waste. This work presents an effective and practical approach for optimizing pyrocarbon sorbent for highly selective gold recovery, and also provides fundamental insights into the mechanism to fully understand the gold recovery process at atomic level.

## Results

### Structural and chemical characterization of pyrocarbon

Pyrocarbon sorbents were synthesized via a straightforward and environmentally friendly method using sodium alginate and calcium chloride as precursors (Fig. 1a). Pyrolysis temperatures were purposefully adjusted to modulate the characteristics of the resulting PyCs ("PyCs" represents the as-synthesized pyrocarbon sorbent pyrolyzed at the temperatures of 500, 600, 700, and 800°C). Scanning electron microscopy (SEM) images exhibited granules at a millimeter scale (-1.0 mm) with flat and smooth surfaces for the PyCs (Supplementary Fig. 1). Transmission electron microscopy (TEM) image indicated that the obtained PyCs possessed a rich porous network structure (Supplementary Fig. 2a, b), corroborated by the Brunauer-Emmett-Teller specific surface area (SSA) analyses (Supplementary Fig. 3a, b). Notably, higher pyrolysis temperatures yielded increased SSAs for the PyCs (ranging from 565.7 to 673.8 m$^2$ g$^{-1}$) while maintaining similar mesoporous structures with pore diameters of 3.79 to 3.97 nm. High-resolution TEM (HRTEM) images revealed the coexistence of conductive graphitic microcrystallites and amorphous carbon within the synthesized PyCs (Supplementary Fig. 2c–f)[40,41]. The observed lattice spacing of approximately 0.35 nm corresponded to the (002) crystal plane of layered graphite carbon (Fig. 1b), which were consistent with X-ray diffraction (XRD) results (Supplementary Fig. 4)[39,42]. To reveal the relative abundance of graphitic and amorphous structures (associated with $sp^2$- and $sp^3$- hybridized carbon), further characterizations were employed to investigate the chemical compositions and structural information of the PyCs. Employing attenuated total reflectance Fourier transform infrared (ATR-FTIR) spectroscopy confirmed notable alterations in functional group chemistry and carbon structure within PyCs as the pyrolysis temperatures increased. (Supplementary Fig. 5a). We identified the formation of phenolic hydroxyl (-OH) and the conversion of aliphatic C-O to aromatic C-O. This was confirmed by the presence of vibrations of aromatic C-OH stretches at 1634 cm$^{-1}$ and the red shift of C-O-C stretches at 1100-1175 cm$^{-1}$. These findings implied an increase of graphitization degree with rising pyrolysis temperatures[35,38,43]. Additionally, the escalating peak intensity at 1545 cm$^{-1}$, associated with aromatic conjugated lattice in graphitic microcrystallites, further supported this result[38]. Furthermore, analysis of Raman spectra indicated an increased $I_D/I_G$ index, signifying heightened defects within the carbon matrices of PyCs at higher pyrolysis temperatures (Supplementary Fig. 5b). This increase in defects was attributed to the intensified condensation of amorphous fragments into graphitic structure by the removal of partial O-containing moieties[38,44]. X-ray photoelectron spectroscopy (XPS) analysis of O 1$s$ and C 1$s$ spectra demonstrated a strong correlation between $sp^2/sp^3$ C and O/C atomic ratios with pyrolysis temperatures in pyrocarbon (Fig. 1c and Supplementary Fig. 6, 7, and Table 1). The thermal decomposition of O-containing groups at high temperature led to a decrease of O/C atomic ratio, indicating a gradually reduction in the oxidation degree from PyC500 to PyC800[38,45]. Interestingly, this trend contrasted to the graphitization degree of PyCs, as evidenced by the change in $sp^2/sp^3$ C atomic ratios[43,46], affirming the growth of graphitic domains in pyrocarbon while depleting amorphous domains as pyrolysis temperatures increased. Moreover, the conductance ($\sigma$) of PyCs was quantified via the Quantum Design physical property measurement system (Supplementary Fig. 8). Evidently, the PyCs exhibited significantly improved $\sigma$ values (from $<2.0 \times 10^{-6}$ to $7.0 \times 10^3$ S m$^{-1}$) at higher pyrolysis temperatures, attributed to the expanding graphite domain in the carbon structure (Fig. 1d). This facilitated electron

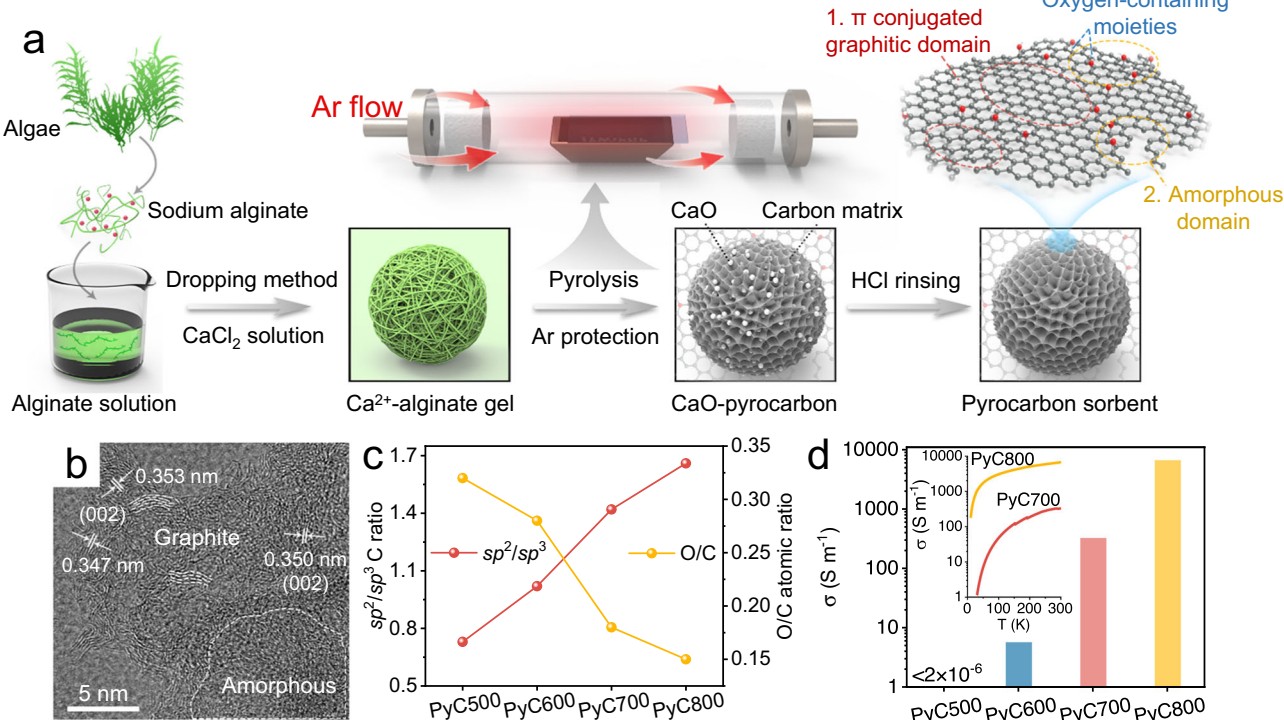

**Fig. 1 | Physicochemical characterizations. a** Scheme of the preparation process of the pyrocarbon sorbent. **b** TEM image of PyC700. **c** XPS-based $sp^2/sp^3$ carbon ratio and O/C atomic ratio of pyrocarbon. **d** The specific conductance of pyrocarbon determined via the Quantum Design physical property measurement system (inset shows the temperature dependence of conductivity in PyC700 and PyC800). Source data are provided as a Source Data file.

transfer within the carbon matrixes, essential in mediating the reduction of gold ions in solution[47,48].

## Highly efficient and selective gold recovery by pyrocarbon

The performance of pyrocarbon sorbent in gold recovery was initially assessed through adsorption isotherm and kinetics experiments. Detailed results of isotherm and kinetics model fitting were provided in Supplementary Table 2, 3 and Fig. 9. Observations revealed that the gold recovery performance of PyCs follows a trend resembling a volcanic curve with increasing pyrolysis temperatures (Fig. 2a). Among its counterparts, PyC700 exhibited the highest theoretical adsorption capacity ($Q_m$) of 2829.7 mg g$^{-1}$ and mass transfer coefficient ($k_f$) of 1.73 × 10$^{-8}$ m s$^{-1}$. Consequently, PyC700 displaying the most promising performance was selected as the optimal sorbent for further investigation across various experimental conditions. At initial concentrations of 10 and 50 mg L$^{-1}$, PyC700 showcased rapid recovery kinetics, achieving equilibrium less than 30 minutes (Fig. 2b), with nearly 100.0% recovery efficiency of Au(III). XRD results indicated the formation of Au nanoparticles (NPs) with major (111) crystal facets (Supplementary Fig. 10)[49], weaking the (002) crystal peaks of the graphite-like structure in pyrocarbon. This suggests the reductive recovery behavior of gold ions by PyC700. Of note, gold recovery capacity significantly impacts the industrial profitability of pyrocarbon-based gold recovery in the acid E-waste leachate. We evaluated the gold recovery capacities across a wide concentration range from 1 to 1000 mg L$^{-1}$ (Fig. 2c), showcasing high recovery capacity of the PyC700 as high as 3000 mg g$^{-1}$ when the Au(III) concentration exceeded 100 mg L$^{-1}$. Within the range from 10 to 100 mg L$^{-1}$, gold capacities of the PyC700 remained between 715.4 and 2346.0 mg g$^{-1}$. The PyC700 also demonstrated high applicability in recycling gold from solutions with low-levels of gold content. The recovery efficiencies surpassed 92.2 and 96.8% at Au(III) concentrations of 50 and 100 μg L$^{-1}$, respectively (Supplementary Fig. 11). These results highlight promising

potential of PyC700 for urban gold mining from waste streams containing diverse gold contents. Temperature-dependent studies conducted at 10, 25, and 60°C revealed a significant influence on gold recovery (Supplementary Fig. 12a). Remarkably, PyC700 achieved an unprecedented $Q_m$ of 6368.4 mg g$^{-1}$ at 60 °C. This exceptional capacity surpasses other carbon-based sorbents (e.g., activated carbon, carbon nanotube, graphene) and advanced nano sorbents (e.g., MOFs, COFs, and metal sulfides), regardless of whether they rely on single sorption force or combined sorption-coupled reduction mechanisms (Supplementary Table 4)[3,15,18,20,50,51]. Thermodynamic calculations based on Van't Hoff equation further confirmed that the pyrocarbon-based gold recovery is an endothermic reaction, supported by the negative standard Gibbs free energy ($\Delta G°$) and positive enthalpy ($\Delta H°$) (Supplementary Fig. 12b and Table 5)[52].

Considering the complexity of actual waste streams, we further assessed the influence of hydrochemical conditions and adsorption selectivity. The gold recovery performance of PyC700 was examined across various acid-alkaline conditions. At an Au(III) concentration of 10 mg L$^{-1}$, the recovery efficiency remained above 99.5% across a broad pH range from pH 1.0 to 8.0 (Supplementary Fig. 13a), reaching as high as 99.9% in acidic HCl solutions of 0.5 and 1 mol L$^{-1}$. At a higher Au(III) concentration of 1000 mg L$^{-1}$, PyC700 exhibited remarkable recovery capacities exceeding 3040.4 mg g$^{-1}$ within the pH range of 3.0 to 8.0 (Fig. 2d). Even in highly acidic circumstances of 0.5 and 1 mol L$^{-1}$ HCl, PyC700 showcased a notable Au(III) recovery capacity exceeding 2801.0 mg g$^{-1}$, highlighting its potential for gold recycling from acidic streams. Overall, PyC700 displayed significantly superior performance under wide acid-alkaline conditions compared to previously reported advanced materials[8,15,53–55], despite limited available data in this context. Accordingly, the Au(III) species always exists as negative ionic forms ($AuCl_4^-$, $Au(OH)_2Cl_2^-$, and $Au(OH)_4^-$) within the pH range of interest[34,56]. The observed trend in capacity variation with pH did not align with the interfacial charges of the sorbent (Supplementary

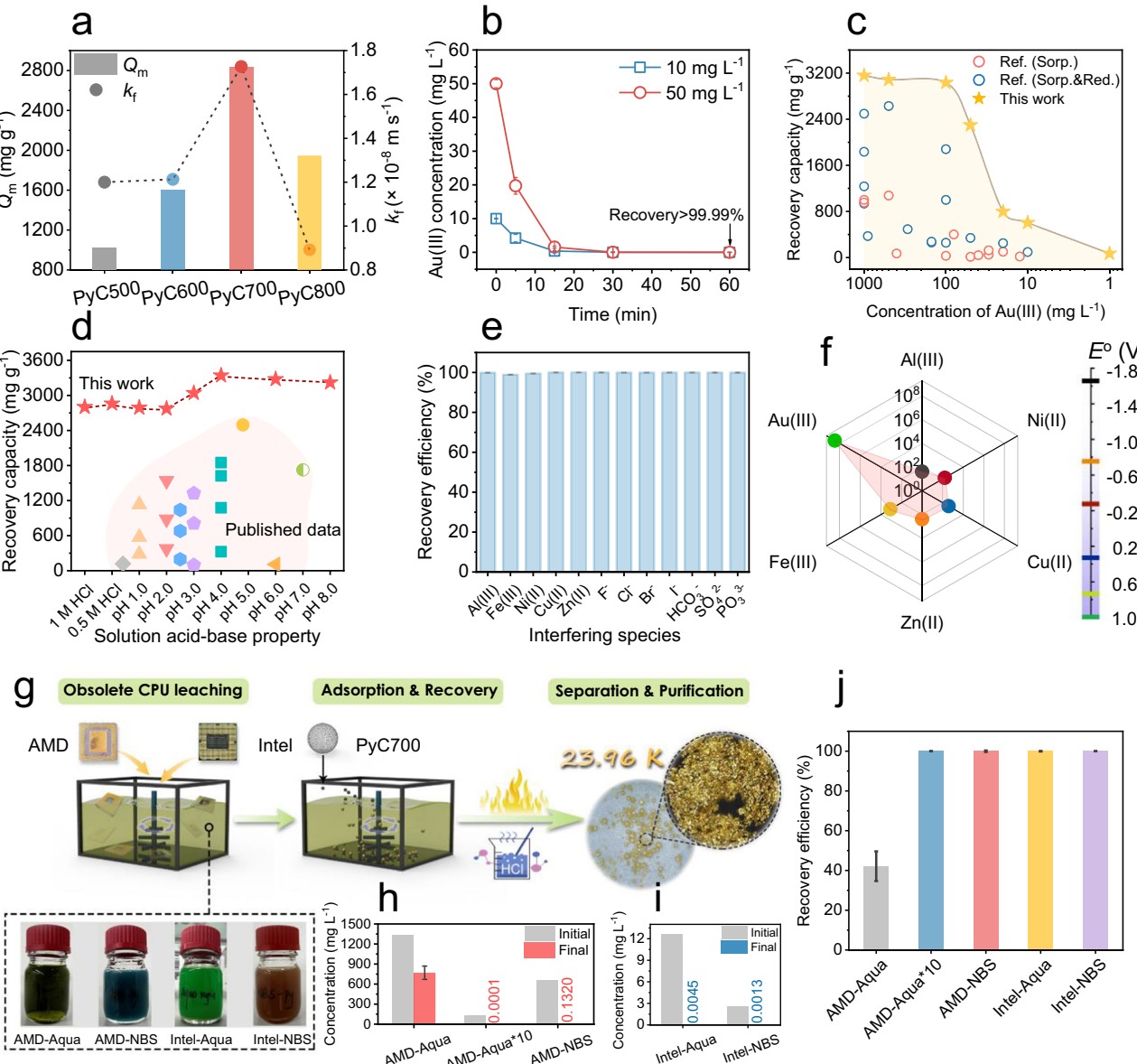

**Fig. 2 | Gold recovery performances. a** The calculated $Q_m$ and $k_f$ of pyrocarbon from Langmuir model and mass transfer model. **b** adsorption kinetics of Au(III) using PyC700. **c** The comparison of Au(III) recovery capacity of PyC700 with that of previously published sorbents. **d** Effects of solution acid-alkaline conditions on Au(III) recovery by PyC700. **e** Effects of potentially interfering species including cations and anions on Au(III) recovery by PyC700. **f** The calculated $K_d$ and the comparison of redox potentials for various metal ions. **g** Schematic of the gold recovery from CPU leachate using PyC700. The concentration change of Au(III) using PyC700 as sorbent in **h** AMD leachate and **i** Intel leachate. **j** The Au(III) recovery by PyC700 in AMD and Intel leachate systems. The abbreviations "Sorp." and "Sorp. & Red." in Fig. 2c and d signify that the reported sorbents functionalized through simple sorption and sorption-coupled reduction processes, respectively. The relevant references for performance comparisons in Fig. 2c and d can be found in Supplementary Table 4. Error bars denote standard deviation of the experiments performed in triplicate. Source data are provided as a Source Data file.

Fig. 13b), emphasizing that chemical reduction plays a dominant role over electrostatic attraction. Additionally, investigations into the effects of coexisting ions (at various concentration (Supplementary Figs. 14 and 15)) on gold recovery verified the exceptional capability of PyC700 to resist interference from both typical cations (Al(III), Fe(III), Ni(II), Cu(II), Zn(II)) and anions (F⁻, Cl⁻, Br⁻, I⁻, HCO₃⁻, SO₄²⁻, and PO₃³⁻) (Fig. 2e). The calculated distribution coefficient ($K_d$) value for Au(III) ($K_d \sim 3.1 \times 10^8$ mL g⁻¹) was approximately 5 to 7 orders of magnitude higher than that for competing metal ions (Fig. 2f), highlighting the remarkable selectivity of PyC700 towards gold ions. The amplitudes of potential fluctuation measured by real-time open circuit potential (OCPT) further suggested significant electron transfer from PyC700 to Au(III) at solid-liquid interfaces (Supplementary Fig. 16a)[34]. In contrast,

no electron transfer occurred in solutions of other cations, except for Fe(III). Potential fluctuation indicated minimal electron transfer between Fe(III) and PyC700 electrode, which was further unveiled by the XRD and XPS results that the Fe(III) was reduced to Fe(II) on the sorbent surface (Supplementary Fig. 16b, c). These findings suggest a reduction potential ($E_0$)-dependent selectivity of PyC700 towards different metal ions (Supplementary Fig. 16d). Considering various leaching agents commonly employed in hydrometallurgy[57], we conducted additional investigations to assess the impacts of prevalent forms of gold complexes, such as AuBr₄⁻ and Au(CN)₂⁻, and Au(S₂O₃)₂³⁻. Our findings revealed a moderate decrease in measured $Q_m$ values by 66.9% for AuBr₄⁻, while significant decreases of 91.1% and 95.7% was observed for Au(S₂O₃)₂³⁻ and Au(CN)₂⁻, respectively

(Supplementary Fig. 17a and Table 6). The hypothesis that adsorption behaviors depend on reduction potential effectively clarifies this phenomenon (Supplementary Fig. 17b, c). It suggests that transferring electrons to $AuBr_4^-$ becomes increasingly difficult, and it's even less probable for $Au(S_2O_3)_2^{3-}$ and $Au(CN)_2^-$ to be reduced to metallic $Au^0$ (Supplementary Fig. 17d). This phenomenon can also be attributed to the differences in the coordination stability constants ($lgK_f^0$) among these three complexes with Au-anions complexes ($lgK_f^0(Au(CN)_2^-)$> $lgK_f^0(Au(S_2O_3)_2^{3-})$> $lgK_f^0(AuBr_4^-)$> $lgK_f^0(AuCl_4^-)$), as higher $lgK_f^0$ value indicates greater difficulty in decomposing the ligands from Au-anions to form metallic Au-Au bonds[58].

## Gold recovery from authentic E-Waste Leachates

The escalating demands for gold prompted our evaluation of PyC700's efficacy in recovering gold from real E-waste leachates. Two types of waste CPUs, Intel CPU and Advanced Micro Devices (AMD) CPU scraps, underwent treatment with two representative chemical leaching systems: aqua regia solution (Aqua) and N-bromosuccinimide (NBS)/ pyridine mixed solution (Fig. 2g)[20]. Among four types of CPU leachates, the AMD-Aqua leachate displayed the highest gold content, with an Au(III) concentration of 1330.1 mg L$^{-1}$ (Figs. 2h, 2i, and Supplementary Table 7). However, the highly acidic nature (pH <0.0) and strong oxidation characteristics of the leachate adversely affected PyC700's gold recovery process, resulting in a recovery efficiency lower than 42.5% ($Q_e$ was 1120.7 mg g$^{-1}$) (Fig. 2j). To address these challenges, we diluted the AMD-Aqua leachate by tenfold (labeled as "AMD-Aqua*10"). As a result, PyC700 reached an adsorption equilibrium less than 300 min and successfully recovered nearly 100.0% of Au(III) in AMD-Aqua*10 leachate (Supplementary Fig. 18), reducing the residual concentration to about 0.1 µg L$^{-1}$, demonstrating profound recovery capabilities in practical applications. Additionally, the resultant AMD-NBS leachate maintained a neutral pH condition and contained lower gold content, with an Au(III) concentration of 650.2 mg L$^{-1}$. In this case, PyC700 exhibited a high recovery efficiency exceeding 99.97%, leading to a residual Au(III) concentration of 0.132 mg L$^{-1}$. In comparison, lower Au(III) concentrations were measured in the Intel scrap leachates (12.65 and 2.60 mg L$^{-1}$ for Intel-Aqua and Intel-NBS, respectively). Following PyC700's gold recovery process, the gold composition in both Intel leachate were almost entirely recovered, with a recovery efficiency exceeding 99.96%. Notably, the near-complete gold recycling from CPUs leachate using PyC700 significantly simplifies the recycling process for other valuable metals (e.g., Cu(II)) present in the remaining solution[37].

For the separation and purification of gold from the spent sorbent, a simple approach combining direct calcination and HCl rinsing was employed. The cost-effective raw materials and facile synthetic protocol ensure the economic viability of the entire gold recovery process, further discussed in the Techno-Economic Analysis (TEA). Gold foil primarily obtained from the calcination of the spent sorbent (Supplementary Fig. 19). Upon HCl rising, optical microscope inspection of the final recycled product revealed minuscule golden particles (Fig. 2g). Despite the coexistence of high concentration of competing metal ions (Fe(III), Cu(II), Co(II), Ni(II)) in leachates, the recycled gold exhibited exceptional purity (>99.82%, 23.96 karat), surpassing that reported in numerous studies[14,37,50,57]. We attribute this high purity to the non-metal nature of pyrocarbon and its remarkable selectivity. The proposed eco-friendly pyrocarbon-based technology with simplified purification processes demonstrates significant applicability in recycling gold from E-waste leachate, advancing the reutilization of non-renewable resources.

## Mechanistic insights into efficient gold recovery
**Underlying mechanism of electron donation and transfer.** Pyrocarbon synthesized at varying pyrolysis temperatures manifests a distinctive volcano-shaped gold adsorption performance closely linked to its physicochemical properties. A trend analysis in Supplementary Method 6 suggested a trade-off among various pyrocarbon properties (e.g., surface functionality, porosity, electrical conductivity, and electron donating capacity) for achieving high gold adsorption efficiency. Among these properties, the reductive activity linked to electron donation plays a crucial role in converting Au(III) to $Au^0$, maintaining a significant Au(III) concentration gradient between the solution and sorbent[37]. Prior researches attributed pyrocarbon's reductive reactivity to intrinsic constituents (i.e., phenolic -OH) and semiquinone-type persistent free radicals[35,43]. However, electron paramagnetic resonance, Boehm titration, and electrochemical analysis revealed that neither intrinsic phenolic -OH groups nor persistent free radicals are the predominant electron donors due to their limited abundance in pristine pyrocarbon (Supplementary Fig. 20 and Table 8). Consequently, the precise mechanism underlying electron donation and transfer for gold recovery on pyrocarbon remains elusive.

To elucidate the electron donation and transfer mechanism, we conducted XPS and *ex*-situ FTIR analyses. XPS analysis demonstrated a gradual increase in O/C atomic ratios of PyC700 after gold adsorption over sequential reaction times (0.1 to 30 min) (Fig. 3a), signifying the oxygenation of carbon networks in PyC700 during gold recovery. Deconvolution of the high-resolution C 1$s$ XPS spectra revealed a 57.8% decrease in $sp^2/sp^3$ C ratios in PyC700 after gold adsorption, indicating the oxidation of graphene-like structures from $sp^2$-hybridized C (e.g., C=C) into $sp^3$ C (e.g., C-C, C-O, and O-C O) (Supplementary Fig. 21 and Table 8). Based on the deconvoluted XPS O 1$s$ spectra and *ex*-situ FTIR results (Supplementary Fig. 22 and 23), an oxidative transformation of π conjugated aromatic structures (graphite domain) into amorphous O-alkylate moieties (amorphous domain) was observed in PyC700 during gold recovery (Supplementary Fig. 24). This suggests that the electrons used for gold reduction were donated from unsaturated $sp^2$-hybridized C with delocalized electrons. To further explore the electron transfer behaviors, in situ FTIR analysis was employed to monitor the interfacial evolution of PyC700. As illustrated in Fig. 3b, the pristine PyC700 exhibited characteristic peaks at 1100 and 1634 cm$^{-1}$, attributed to vibrations of aromatic C-O/C-O-C[38] and phenolic -OH[43], respectively. The peak at 3442 cm$^{-1}$ corresponded to the stretching vibration of O-H bonds in the adsorbed water molecules ($H_2O_{ad}$)[59]. In situ FTIR spectra and corresponding heat mappings revealed a simultaneous increase in the peak intensity of phenolic -OH and $H_2O_{ad}$ molecules as the reaction initiated (Fig. 3b and Supplementary Fig. 25), suggesting the water molecules adsorption and the generation of phenolic -OH over PyC700. The synchronous peak intensity changes of phenolic -OH and $H_2O_{ad}$ throughout the gold recovery process implied that the $H_2O_{ad}$ molecules significantly contributed as oxygen sources for the generation of phenolic -OH groups on pyrocarbon structures (i.e., hydroxylation). Control experiments further supported this speculation, revealing that the oxygen used for pyrocarbon oxygenation originated from $H_2O$ molecules in water matrices rather than dissolved oxygen (Supplementary Fig. 26). The interfacial electron states and dehydrogenation ability of hydroxyl-anchored aromatic structures were further investigated using Bader charge analyses. Other O-containing groups such as aldehyde and carboxyl were also considered for reducibility comparison. Among all counterparts, types of phenolic -OH groups located at zigzag edges and within the graphitic plane exhibit the highest propensity for proton loss, as evidenced Fig. 3c. Additionally, the distinctive Bader charge values ($\Delta q$) indicate different electron delocalization behaviors of these two types of phenolic -OH groups (Fig. 3d and Supplementary Fig. 27). The high $\Delta q$ value with zigzag-edged phenolic -OH suggests high electron cloud density on the oxygen atoms, facilitating gold ion capture and reduction at this site. Conversely, the low $\Delta q$ value for in-plane phenolic -OH indicates that the electron cloud density was biased towards aromatic structures from oxygen atoms, signifying the

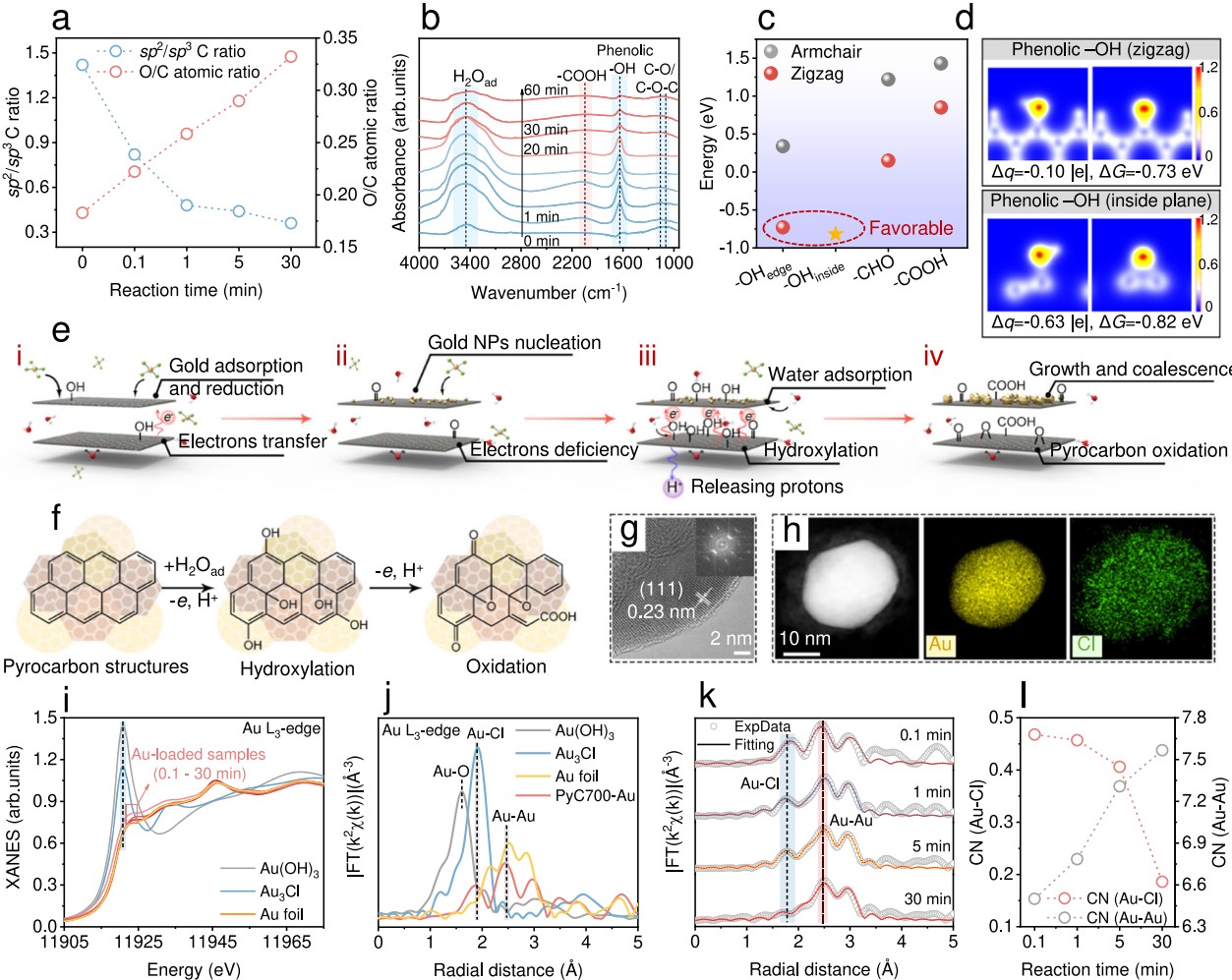

**Fig. 3 | Characterizations for mechanistic investigation. a** XPS-based $sp^2/sp^3$ carbon ratio and O/C atomic ratio of PyC700 for different adsorption durations. **b** In situ FTIR spectra of PyC700 during gold recovery. **c** Bader charge analysis of different O-containing groups in pyrocarbon. (**d**) ELF slice maps of phenolic hydroxyl groups located at the zigzag-edge and within the graphene plane. **e** Schematic illustration for the gold recovery on pyrocarbon. **f** Schematic of the hydroxylation of aromatic structures and their oxidation transformation process.

**g** HRTEM image of the gold NPs on pyrocarbon. **h** HAADF-STEM of gold NPs (EDS mapping images of Au and Cl elements were provided along with the HAADF-STEM image). **i** Au $L_3$-edge XANES spectra of Au-loaded PyC700. **j** FT of $k^2$-weighted EXAFS spectra of Au-loaded PyC700. **k** FT of $k^2$-weighted EXAFS spectra of Au-loaded PyC700 with different reaction durations. **l** The evolution of CNs of Au-Cl and Au-Au interactions in Au-loaded PyC700 with different reaction durations. Source data are provided as a Source Data file.

potential to transfer its electrons for gold ions reduction through the conductive carbon matrices.

Based on the preceding results, we proposed a plausible mechanism concerning the electron donation and transfer on pyrocarbon, outlined in Fig. 3e: (i) A limited number of electrons are transferred from intrinsic phenolic -OH groups on pyrocarbon to reduce the adsorbed gold ions to $Au^0$. (ii) As the electron supply diminishes, gold ions continue to be adsorbed on the solid-liquid interfaces, causing a significant electron deficiency due to accumulated excess gold ions. (iii) Surface-sorbed $H_2O$ molecules undergo dissociation into hydroxyl ions and protons catalyzed by $Au^0$ NPs as reported in previous studies[60–62]. Subsequently, hydroxyl ions induce the hydroxylation of aromatic structures, generating phenolic -OH groups on unsaturated $sp^2$-hybridized C (at the edge or inside the plane of graphitic domains) (Fig. 3f). (iv) Both the hydroxylation process and the oxidation of phenolic -OH contribute electrons to the gold recovery, releasing protons into solution systems (Supplementary Fig. 28). These contributed electrons facilitate the reduction of gold ions on conductive carbon matrices, consequently oxidizing pyrocarbon containing phenolic -OH groups into highly oxidized carboxyl, quinone, or alkyl ether moieties.

**Understanding the gold reduction mechanism.** To gain a comprehensive understanding of the gold recovery mechanism, we further investigated interfacial gold reduction processes over pyrocarbon. Time-resolved SEM images showed the primary formation of gold seeds with diameters around 20 nm on the pyrocarbon surface (Supplementary Fig. 29). Notably, these gold seeds exhibit a significantly high surface free energy[34,63], rendering them inherently unstable in a state of boiling that facilitates the consumption of Au(III) salt. Simultaneously, the enhanced PyC700 conductivity further promotes efficient electron transfer for subsequent reduction of gold ions (Supplementary Fig. 30). As gold salt is consumed at localized adsorption domains, bare $Au^0$ NPs continuously attract gold ions from solution for self-growth. This phenomenon can be perceived as an autocatalytic surface growth process that promotes overall gold recovery process on pyrocarbon[64,65]. By following Ostwald ripening rules[66], $Au^0$ NPs gradually increase in size until reaching approximately 200 nm, following which NPs coalescence leads to the formation of aggregates of $Au^0$ NPs. The HRTEM image (Fig. 3g) revealed the presence of (111) crystal planes in $Au^0$ NPs, exhibiting an interlayer spacing of 0.23 nm, consistent with the XRD results shown in Supplementary Fig. 10. TEM images revealed the presence of amorphous structures

surrounding the newly formed $Au^0$ NPs (Supplementary Fig. 31), which disappeared after a 5-minute adsorption process. High-angle annular dark-field scanning transmission electron microscopy (HAADF-STEM) and element mapping images confirmed a close proximity between Au and Cl elements (Fig. 3h), with Cl extending more towards the periphery of the amorphous structures. As the gold reduction reaction progressed, mapping images of larger $Au^0$ NPs exhibited complete overlap between these two elements (Supplementary Fig. 32). These observations strongly indicate the existence of gold-chlorine complexes that envelop gold nuclei during the gold reduction process.

To elucidate the gold reduction mechanism at atomic level, we explored the time-resolved evolutions of chemical state and local coordination environments of Au in Au-loaded PyC700 using Au $L_3$-edge X-ray absorption near-edge structure (XANES) and extended X-ray absorption fine structure (EXAFS) spectra with three reference standards. The XANES spectra (Fig. 3i) indicated a higher intensity of normalized white line peak (11921 eV) of the Au-loaded PyC700 compared with that of Au foil[67], suggesting the existence of unreduced $Au^{x+}$ (x = 0–3) species in the initially nucleated $Au^0$ NPs. Moreover, the white line peak intensities in Au-loaded PyC700 eventually became almost identical to that observed for Au foil, showcasing the complete reduction of gold ions. Linear combination fitting (LCF) of the Au-loaded samples was conducted to gain deeper insights into the changes observed in XANES spectra (Supplementary Fig. 33). LCF analysis confirmed that a portion of Au in the Au-loaded PyC700 existed in a cationic state during the initial stage[68], with the average oxidation state of Au(+0.33) (referenced to $Au_3Cl$ standards) constituting for 17.3 and 7.9% at 0.1 and 1 min, respectively. As the reaction progressed, no contribution from Au(+0.33) could be detected due to low proportion of unreduced $Au^{x+}$ (x = 0–3) on the large-sized Au NPs. Fourier-transformed (FT) EXAFS spectroscopy was employed to identify the localized structure of Au within short range (<4 Å). The curve fitting results for the samples were displayed in Supplementary Fig. 34, Fig. 35 and Table 9. The R space spectrum of Au-loaded PyC700 (Fig. 3j) revealed distinct peaks corresponding to Au-Cl and Au-Au scattering at distances approximately 2.27 and 2.86 Å, respectively, referenced by Au foil and $Au_3Cl$ standards[69]. These findings were consistent with time-resolved FT-EXAFS spectra and coordination number (CN) analysis of Au-loaded PyC700, indicating a decrease in the proportion of Au-Cl interactions and an increase in Au-Au interactions within local gold coordination environments during gold recovery (Figs. 3k, l). Consequently, XANES and LCF analyses combined with EXAFS findings, collectively confirmed the presence of gold-chlorine clusters during initial stages of gold nucleation and growth[70]. Rapid growth of Au NPs to large size leads to negligible presence of Au-Cl interactions among gold coordination structures.

To comprehend the adsorption and reduction processes of gold ions by pyrocarbon, spin-polarized density functional theory (DFT) calculations were performed. Constructed graphene supercells functionalized with hydroxyl groups were employed to evaluate the adsorption energy ($E_{ad}$) of $HAuCl_4$ molecules, which were sequentially adsorbed at hydroxyl sites on the graphene structure. The entire gold reaction pathway and its corresponding free energy diagram were summarized in Fig. 4a, b, with detailed step-by-step descriptions (i.e., adsorption and dechlorination process) available in the Supplementary Information. Specifically, the DFT calculations unveiled a favorable adsorption of $HAuCl_4$ molecules onto pyrocarbon (referred to as "$G^*HAuCl_4$") with an $E_{ad}$ of −1.45 eV (Supplementary Fig. 36). The adsorbed $HAuCl_4$ subsequently underwent successive dechlorination reactions, releasing three HCl to create the $G^*AuCl$ intermediate, wherein the Au atom directly bonded to an O atom within hydroxyl group. Following the classical nucleation theory, recent studies suggested that direct detachment of all four Cl atoms from $HAuCl_4$ leads to the formation of $Au^0$[14,53]. Our findings indicated a positively unfavorable $E_{ad}$ value (1.08 eV) for this process on pyrocarbon, suggesting

the presence of a high energy barrier during the detachment of the last Cl atom from $HAuCl_4$ to form $Au^0$ nuclei. This elevated energy barrier has been similarly documented in previous studies regarding the dechlorination from Au-Cl intermediates[67]. Instead, we found that the adsorption of an additional $AuCl_3$ on the $G^*AuCl$ intermediate was found to be energetically favorable with an $E_{ad}$ of −2.39 eV ($AuCl_3$ was used instead of $HAuCl_4$ in subsequent calculations due to facile HCl detachment from $HAuCl_4$). Subsequent sequential adsorption processes of $AuCl_3$ molecules onto $G^*Au_iCl_j$ intermediates, accompanied by stepwise dichlorination, occurred with continuous exothermic behaviors (Supplementary Fig. 37 to 49). These processes continued until the formation of $Au^0$, which were bound with six other Au atoms within the $Au_{15}Cl_{14}$ clusters ($G^*Au_{15}Cl_{14}$) (Supplementary Fig. 50). This specific structure is termed a ligand-protected metal cluster, aligned with the 'divide-and-protect' concept, theoretically displaying lower energy level compared to typical Au-Cl linkages[71]. Based on DFT calculations, the reduction of gold ions is proposed to occur through the stepwise nucleation of chlorine-protected gold clusters, which is crucial in gradually lowering the energy barriers for the entire reduction processes. These findings elucidate the experimentally observed inverse trend of CNs for Au-Cl and Au-Au paths, as well as the homogeneous distribution of Au and Cl elements on the pyrocarbon. In summary, we systematically unraveled interfacial behaviors involving electron donation and transfer, as well as gold nucleation and gradual growth on pyrocarbon surfaces, as summarized in the proposed gold reduction mechanisms in Fig. 4c.

## Discussion

To evaluate the economic feasibility of the pyrocarbon adsorption method for recovering gold from E-waste leachate, a thorough technoeconomic analysis (TEA) centered on the specific material flow for gold recovery using PyC700 was conducted (Supplementary Fig. 51). The determination of technological parameters was extensively discussed in Supplementary Method 7. Initially, the cost breakdown was depicted in Fig. 5a and detailed in Supplementary Table 10. This breakdown encompassed CPU waste leaching, sorbent production, gold recovery, and purification steps. The Sankey diagram (Fig. 5a) illustrates that the largest share of the material cost is linked to CPU purchase, accounting for 65.04%, followed by labor costs for employees at 15.32%. Because of the straightforward and cost-effective synthesis method for pyrocarbon, the total capital and material expenses for sorbent production were approximately 10.38%. According to the TEA findings, the annual income was calculated to be US$1.2 × 10^6 (Fig. 5b), markedly exceeding the total annual input cost of US$8.7 × 10^4. Hence, employing pyrocarbon-based adsorption for gold recovery from CPU waste can yield an impressive input-output ratio of 1370%, underscoring its significant economic promise. As indicated in the sensitivity analysis (Fig. 5c), the potential economic efficiency of this method could be theoretically enhanced by reducing labor costs and identifying more cost-effective pathways for recycling CPU waste. However, these approaches pose challenges due to the unregulated E-waste recycling market and the complexities associated with establishing cross-border factories. Conversely, increasing the adsorption temperature from 25 to 60 °C notably reduces the specific cost by 19.2% (Fig. 5c, Supplementary Fig. 52 and Table 11). This translates to an input-output ratio of 1692%, emphasizing the importance of enhancing the gold recovery capacity of the sorbent.

In conclusion, our proposed pyrocarbon-sorption technique offers a practical scenario for recovering gold from waste streams, and provides an economically viable capital option with environmentally sustainable costs. The developed pyrocarbon demonstrates remarkable efficiency as a gold sorbent, exhibiting a high recovery capacity (2829.7 mg g$^{-1}$), exceptional gold selectivity (with a $K_d$ of 3.1 × 10$^8$ mL g$^{-1}$) and remarkable adaptability to a broad spectrum of environmentally relevant conditions (such as varying

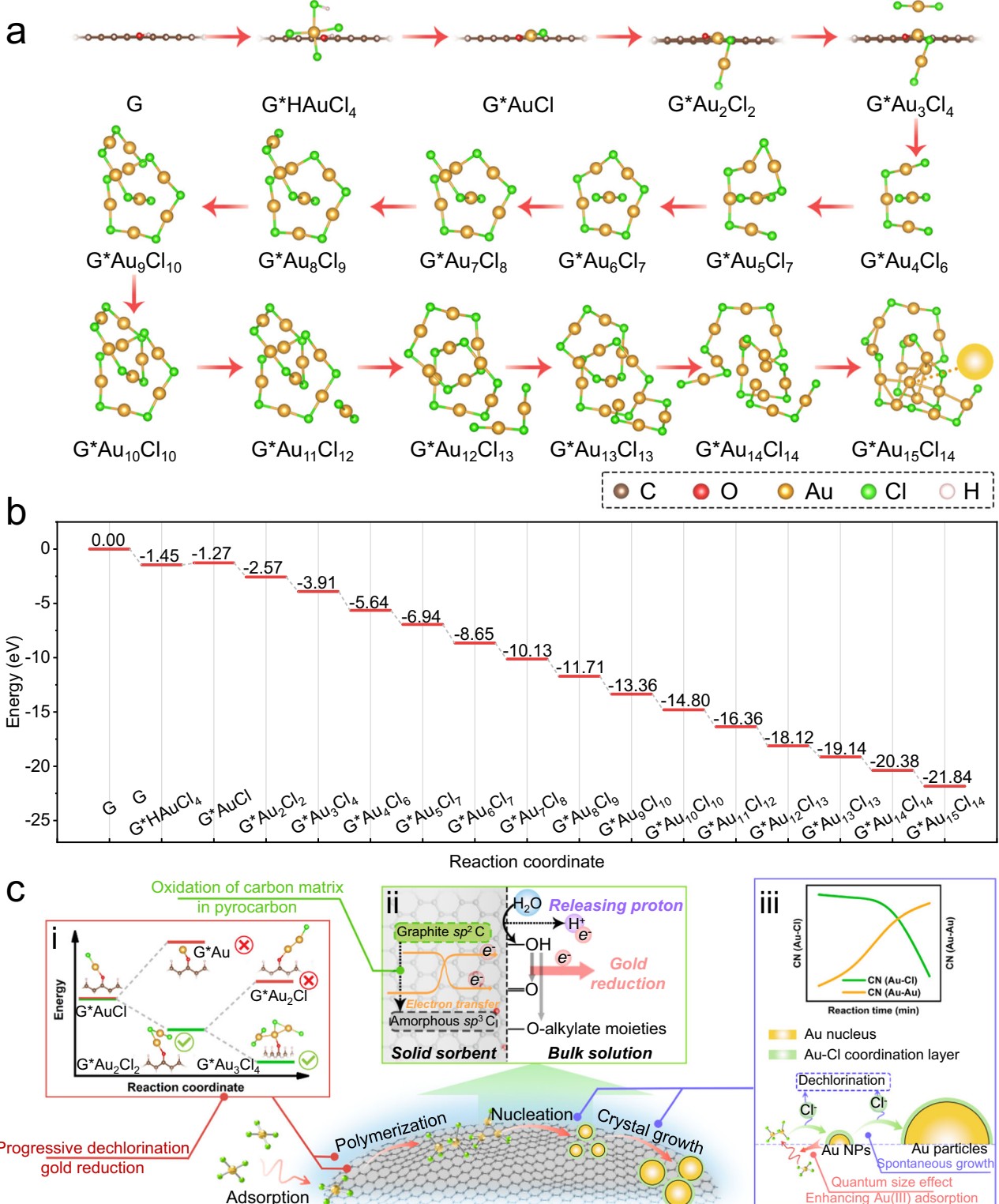

**Fig. 4 | Theoretical calculations for mechanistic investigation. a** Schematic illustration of the proposed mechanism of gold reduction processes on pyrocarbon. **b** The free energy diagram of the proposed gold-chlorine intermediates in gold reduction processes. **c** Overview of the intrinsic principle of selective and efficient gold recovery performance of pyrocarbon-based sorption technique.

pH and ionic strength). Furthermore, this pyrocarbon efficiently and selectively extracts gold from actual CPUs leachates, ensuring high gold purity (>99.82%, 23.96 karat) and facilitating the sustainable reutilization of end-of-life electronic devices. The technoeconomic analysis confirms the substantial economic potential of the

aforementioned recovery process, showcasing an impressive input-output ratio of 1370%. In-depth characterizations and DFT calculations further elucidate the pivotal role played by the unsaturated pyrocarbon aromatic structures in transferring their delocalized π-electrons for gold reduction. This study sheds light on an

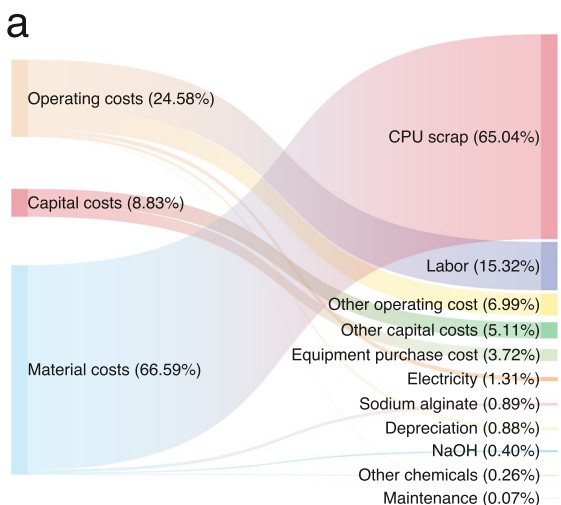

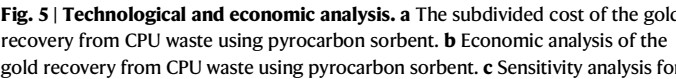

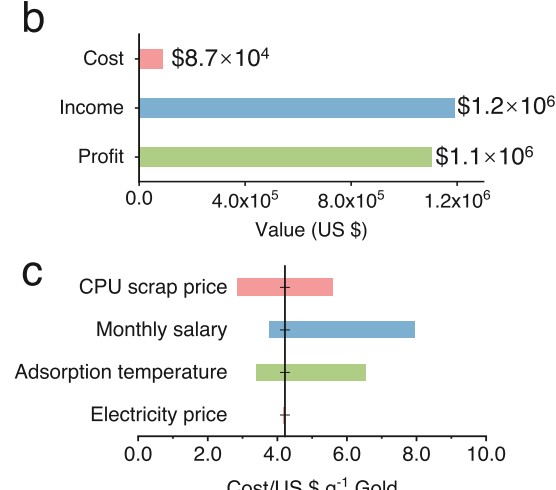

**Fig. 5 | Technological and economic analysis. a** The subdivided cost of the gold recovery from CPU waste using pyrocarbon sorbent. **b** Economic analysis of the gold recovery from CPU waste using pyrocarbon sorbent. **c** Sensitivity analysis for the gold recovery from CPU waste using pyrocarbon sorbent (the assumed values are provided in Supplementary Table 11). Source data are provided as a Source Data file.

effective approach to harnessing eco-friendly biomass for the preparation of high-performance sorbents, achieved through the customization of its surface functionality and structures. We anticipate that this sorption technique will find widespread adoption for low-cost, environmentally friendly, and selective extraction of non-renewable gold metal resources, enabling sustainable urban mining and other pertinent applications.

## Methods
Experimental details on the sorbent preparation, characterization, performance, mechanism investigation procedures, electrochemical analysis, and techno-economic analysis are presented below.

### Preparation of pyrocarbon
Pyrocarbon sorbents were prepared using a sequential approach involving continuous dropping gelation, inert pyrolysis, and acid rinsing. Under stirring at 60 °C, 20.0 g of SA was dissolved in 2.0 L deionized water to yield a yellowish transparent SA solution (1.0 wt%), which was then continuously pumped as droplets into a 4.0 L CaCl$_2$ solution (2.0 wt%) at a flow rate of 0.3 L h$^{-1}$. The resulting calcium alginate gels were cured in the CaCl$_2$ solution for 12 h and subsequently freeze-dried to obtain calcium alginate granules. These granules were pre-carbonized at 300 °C for 2 h, followed by pyrolysis at various temperatures (500, 600, 700, and 800 °C) for an additional 2 h with a heating rate of 5 °C min$^{-1}$ under an argon atmosphere. Next, the collected pyrocarbon was washed with 2 M HCl and deionized water to remove inorganic calcium components. Finally, the pyrocarbon granules were freezing dried to obtain the resultant pyrocarbon sorbents (e.g., PyCs).

### Analysis and characterization
TEM images were measured using a transmission electron microscope (TEM, Hitachi JEM-3010) at an accelerating voltage of up to 200 kV. The chemical information of the pyrocarbon samples was analyzed via XPS (EscaLab Xi+, Thermo Fisher Scientific) with an Al-Kα X-ray source. The electrical transport properties of the pyrocarbon were determined via the Quantum Design physical property measurement system (PPMS), in which samples was exfoliated into pieces for carefully analysis of specific conductance. In situ FTIR of the pyrocarbon reaction in Au(III) solution was recorded in a Nicolet 6700 FTIR spectrometer (Thermo Fisher Scientific, USA) equipped with a MCT detector. The X-ray absorption fine structure

spectra (XAS) (Au L$_3$-edge) of the Au-loaded samples (prepared via pre-adsorption of Au(III) onto PyC700 at sequential reaction time) were collected at XAS Beamline at the Australian Synchrotron in Melbourne, Australia, using a set of liquid nitrogen cooled Si(111) monochromator crystals. The electron beam energy is 3.0 GeV. With the associated beamline optics (Si-coated collimating mirror and Rh-coated focusing mirror), the harmonic content of the incident X-ray beam was negligible. Data was collected by using transmission mode, and the energy was calibrated using an Au foil. The beam size was about 1 × 1 mm. Note that a single XAS scan took ~1 h. The Demeter Athena and Artemis program was adopted for XANES, LCF and EXAFS data analysis. The concentration of metal ions in the filtration solutions were determined using inductively coupled plasma mass spectrometer (ICP-MS, Agilent 8900 series, Agilent Technologies). More information of analysis and characterization of the materials was provided in Supplementary Method 2.

### Gold recovery performance investigation
The stock Au(III) solution was prepared by dissolving Gold(III) chloride (HAuCl$_4$) in deionized water. Various initial concentrations of the Au(III) solution in batch experiments were prepared by diluting the stock Au(III) solution. Kinetics tests were operated at 10 and 50 mg L$^{-1}$. Thermodynamic isotherm adsorption experiments were performed by adjusting reaction temperatures (10, 25, and 60 °C) in thermostatic oscillator. The solution acid-base property in pH influence tests was adjusted via dropwise addition of 0.5 mol L$^{-1}$ HCl or NaOH to the solution (solution containing 0.5 and 1.0 mol L$^{-1}$ HCl were prepared by diluting concentrated HCl). The impacts of competing cations and anions were investigated by adding certain amount of high concentration salts solution into Au(III) solution. Based on the typical components in E-wastes[16,72], Al(III), Fe(III), Ni(II), Cu(II), and Zn(II) were selected as representative competing cations in the selectivity studies. To identify the influence of Au(III) species, auric cyanate (Au(CN)$_2^-$) and tetrabromoauric acid (AuBr$_4^-$) solution with varying concentration was also employed in isotherm tests. All batch experiments were operated at constant agitation (180 rpm) in thermostatic oscillator to reduce mass transfer resistance. Unless otherwise stated, initial solution pH was about 3.0, reaction time was 12 h, sorbent dosage was 0.5 g L$^{-1}$, and solution volume was 20 mL, in all the experiments. After an incubation period of 12 h to reach equilibrium, the residual solution was filtered and diluted for ICP-MS measurement. The graphs with error

bars are presented with standard deviation of the experiment in triplicate.

The recovery capacity ($q_e$), the recovery efficiency (%) and the distribution coefficient ($K_d$) were calculated using equations as follows:

$$q_e = \frac{V(C_0 - C_e)}{m} \tag{1}$$

$$\text{Recovery efficiency} = \frac{(C_0 - C_e)}{C_0} \times 100\% \tag{2}$$

$$K_d = \frac{(C_0 - C_e)}{C_e} \times \frac{V}{m} \tag{3}$$

where $V$ is the volume of the Au(III) solution (mL), $m$ is the mass of the sorbent (g), and $C_O$ and $C_e$ are the initial and equilibrium solution concentrations of Au(III) (mg L$^{-1}$), respectively. The adsorption models (e.g., Langmuir and mass transfer models) and thermodynamic calculations were provided in Supplementary Method 3.

### Gold recovery from CPU waste leachates

Prior to leaching, the two types of CPU boards were crushed into scrap (>20 mesh). Two types of leaching agents including aqua regia and N-Bromo succinimide-pyridine (NBS-Py) systems were employed to prepare Au(III)-containing leachate (Details were presented in Supplementary Method 4). Based on the respective preparation procedures, the obtained CPU leaching solution were labeled as AMD-Aqua, AMD-Aqua*10 (prepared by diluting AMD-Aqua 10 times), AMD-NBS, Intel-Aqua, and Intel-NBS, respectively. For Au(III) recovery from leachate, 0.05 g of PyC700 was added into 50 mL leachate, which was subsequently agitated at 180 rpm and maintained at 25 °C. After an incubation period of 12 h to reach equilibrium, the residual solution was filtered and diluted for ICP-MS measurement.

### Gold separation and purification

The purification of gold was achieved through the combination of calcination and acid treatment strategy. 100 mg of the Au-loaded PyC700 (prepared in AMD-Aqua solution) was loaded in a tube furnace and heated to 900 °C under air with heating rate of 10 °C min$^{-1}$. The temperature was held for 2 hours and then allowed to cool to room temperature. Subsequently, the obtained reddish-brown powder was dispersed in 10 mL concentrated HCl and stirred for 6 h. The acid rinsing process was repeated once more to dissolve any possible metals on the gold product. Finally, the golden Au powders were separated from the clear acidic solution and washed with deionized water for 3 times prior to purity test. Specifically, the purity of the recovered gold was tested referring to the standard GB/T 38145-2019.

### Electrochemical analysis tests

All electrochemical measurements were performed in a three-electrode configured cell using an electrochemical workstation (CHI 760e, Shanghai Chenhua Instrument Co., LTD). The pyrocarbon drop-coated glassy carbon electrode and glassy carbon cylinder were used as working electrodes in OCPT measurements and electron accepting capacity/electron donating capacity tests. Ag/AgCl (saturated KCl) and platinum electrodes were used as reference and counter electrodes, respectively. Detailed experimental processes for electrochemical analysis tests were described in Supplementary Method 5.

### Theoretical calculations

Theoretical calculations were performed based on DFT calculations as implemented in the Vienna Ab initio Simulation Package (VASP, version 5. 4. 4). The Perdew-Burke-Ernzerhof functional with generalized gradient approximation was employed for exchange correlation[73]. To evaluate the adsorption of Au(III) molecules on pyrocarbon, 5 × 5 × 1 supercells of graphene with hydroxyl groups were constructed. HAuCl$_4$ were utilized to represent the +3-charge state of Au in adsorption of first Au, and subsequently AuCl$_3$ was used due to the facile HCl detachment from HAuCl$_4$ as observed by the favorable free energy change. A vacuum layer of at least 15 Å was included to prevent spurious interactions among periodic images. During the optimization, the tolerances of energy and force were $2 \times 10^{-5}$ Ha and 0.004 Ha/Å, and the maximum displacement was $5 \times 10^{-3}$ Å, respectively. The dispersion-corrected DFT (referred as DFT-D3) was utilized to account for the van der Waals interactions to evaluate the ligand effect[74]. The adsorption energy ($E_{ad}$) was defined as $E_{ad} = E_T - E_G - E_{Au}$, where $E_T$, $E_G$, and $E_{Au}$ represent the total energy of the adsorption system, the graphene, and Au(III) species, respectively. The reaction free energy of releasing protons and electrons in our calculation was computed by referencing the equation: $G_{proton+electron} = G_{1/2H2(g)} - neU$ as recommended by Norskov and Bockris[75,76], where $e$ and $U$ represent the transfer electron and electrochemical potential, respectively. The crystal data of optimized computational models have been provided in the Source Data file.

### Techno-economic analysis

Referencing to the Guidelines for Techno-Economic Analysis of Adsorption Processes[77], the production step of PyC700 could meet the industrial suitability, taking into account the required characteristics (e.g., water tolerance, rapid mass transfer, low cost, low environmental and safety impact). Techno-economic analysis was employed to evaluate the economic potential of this pyrocarbon-based gold recovery process. Details of the TEA methodologies, and the specific sample for the calculation of total cost of the proposed gold recovery process from CPU scrap using pyrolytic PyC700 could be found in Supplementary Method 7.

### Reporting summary

Further information on research design is available in the Nature Portfolio Reporting Summary linked to this article.

## Data availability

The data supporting the findings of the study are available within the paper and its Supplementary Information. Source data are provided with this paper.

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

## Acknowledgements

The work was supported by the National Natural Science Foundation of China (22176124, J. L.), the China Postdoctoral Science Foundation (2023M742246, K. F.), and the Research Start-up Funding from Shanghai Jiao Tong University (WH220416002, J. L.). The authors appreciate the help for Quantum Design physical property measurement from Dr. Guoxiong Tang and Prof. Hui Xing from Key Laboratory of Artificial Structures and Quantum Control, and Shanghai Center for Complex Physics, School of Physics and Astronomy, Shanghai Jiao Tong University. The authors also thank the help from Shiyanjia Lab (www.shiyanjia.com) for the XAFS characterizations. The views and ideas expressed herein are solely of the authors and do not represent the ideas of the funding agencies in any form.

## Author contributions

K.F. and J.L. developed the concept, and designed the research. K.F. prepared the materials, and carried out the experiments and characterizations. X.L., Y.P. and K.F. planned and carried out the DFT calculations. K.F., X.L., N.Z., X.Z., S.Z. and J.L. contributed to the interpretation of the results. K.F. wrote the manuscript. J.L., Y.P., X.Z. and N.Z. revised the manuscript. All authors provided critical feedback and helped shape the research, analysis, and manuscript.

## Competing interests

The authors declare no competing interest.
