## [Peer Review File · Nature Communications]

Utilizing Cost-Effective Pyrocarbon for Highly Efficient Gold Retrieval from E-Waste LeachateREVIEWER COMMENTS

Reviewer #1 (Remarks to the Author):

The manuscript (MS) presents the results from a series of experiments studying the synthesis and gold recovery behaviors of pyrocarbon sorbent, evaluated its potential in gold recycling from real electronic waste devices (CPUs), and investigated the internal sorption mechanisms using in situ FTIR, XPS, synchrotron XAS and DFT calculations. Compared with most previously reported gold sorbents, the developed pyrocarbon exhibited transcendental gold recovery performance, for its fordable production costs, satisfactory capacity, and easily separated properties. The Au(III) adsorption-reduction- nucleation processes were elucidated, in which the aromatic region in carbon matrices was proved play crucial role in forming reductive phenolic structures, and water molecules were found to participate in sorption processes; this is interesting and can enlighten us to pay more attention in chemical reactions in sorption. Through the specific material flow and cost calculations, the techno-economic analyses were employed to verify the cost-effectiveness of carbon-based gold sorption process. This paper offers a new and economically viable solution for gold recycling from aqueous solution. The experiments are reasonably well designed, and the mechanism interpretation are mostly supported by the data. In structural view, there are some points that need to be taken into consideration. Overall, it thinks this work is suitable for publication after minor revision considering following comments.

Special Comments:

1. Pore diffusion plays a crucial role in sorption processes, particularly for the sorption of metal ions on char granules investigated in this study. BET analyses revealed an increase in specific surface areas for pyrocarbons and provided information on mesopores; however, micro pores with diameters less than 2 nm were not quantified (Supplementary Fig. 3b). It is imperative to provide more comprehensive details regarding the internal pore structure encompassing a wider range of pore sizes.
2. Except for the abbreviations used for characterizations (e.g., XPS, HRTEM), there are numerous other abbreviations in the manuscript, such as BCTs, BT500-800, PTs, Ph-OH, etc. It is essential to ensure that these abbreviations are necessary and do not confuse the readers.
3. The authors employed the term "pyrocarbon" to describe the prepared sorbent, while using the abbreviation "BC" for brevity. It is speculated that "BC" likely refers to biochar. Please ensure consistency in abbreviations with their original meanings.
4. There are some written or illustrated details that require revision.: (1) Supplementary Fig. 23 should provide the full name of GRP for clarity; (2) In MS, it was stated that the purity of recycled gold from CPUs was 23.96 karat; however, this contrasts with the value shown in Fig. 2g (23.9 karat); (3) Line 859: Please correct "Nature Sustainability" to "Nature Communications". Please thoroughly review these points and meticulously examine the entire manuscript to prevent the occurrence of such errors.
5. In this work, the authors employed mass transfer coefficient k_f to distinguish the adsorption kinetics of those BCTs sorbents, however it suggests kinetic constants were used more frequently in fitting kinetics data. What are the advantages of using k_f compared to typical kinetic constants?

6.The OCPT depicted in Supplementary Fig. 16a suggests a recovery selectivity that is dependent on the reduction potential. This observation indicates the occurrence of electron transfer for Fe(III). What are the reduction products of Fe(III) ions? Could the presence of Fe(III) hinder the reduction of gold ions due to competition for electrons?

7.The adsorption capacities for AuBr₄⁻ and Au(CN)₂⁻ exhibited a significant decrease, necessitating further discussion to elucidate the underlying reasons for this phenomenon. Additionally, it is imperative to clarify the adsorptive and/or reductive forms of these two Au species.

8.The cations of Al(III), Fe(III), Ni(II), Cu(II), Zn(II) were used to investigate the adsorption selectivity. It is essential to clarify the selection criteria for these cations in the MS. Moreover, it is crucial to determine whether these cations are present in the actual E-waste leachates prepared in this study.

9.For the techno-economic analyses of sorbent production, more details regarding the specific material flow should be provided, such as material ratios, reaction parameters, etc. Additionally, it is necessary to clarify price parameters (e.g., electricity costs, manufacturing device expenses) that may impact the overall costs.

Reviewer #2 (Remarks to the Author):

The work describes the design and mechanistic study of pyrocarbons for the recovery of gold. The study describes the nucleation of gold on the functional groups at the surface of the carbon, and a combined DFT study to explain the mechanism. Finally, a TEA and proof-of-concept for real-world CPU is used.

While the work is certainly of technical interest, and the redox-transformation of gold brings significant benefits, the overall novelty of the work is not considered sufficient for Nature Communications.

Activated carbon and other forms of carbons that contain aromaticity and phenolic groups have been extensively used as traditional methods for gold adsorption, including at an industrial scale, and while there are interesting mechanistic insights on the particular carbon chemistry shown in this work, the framework for separation remains known. Therefore, the work is of more specialized and incremental technical interest than transformative.

Therefore, a more specialized journal in sustainability or separation technology is recommended for the work.

- A major technical comment is that a comparison between surface adsorption type mechanisms, as shown for many of the published data in 2d, and a gold reduction process, may be an unfair comparison as nucleation can give higher adsorption capacities. A more relevant comparison would be with other reduction methods (e.g. chemical reduction or electro-deposition), due to the

possibility of multi-layer and higher overall areal capacity.

Reviewer #1

General comment: *The manuscript (MS) presents the results from a series of experiments studying the synthesis and gold recovery behaviors of pyrocarbon sorbent, evaluated its potential in gold recycling from real electronic waste devices (CPUs), and investigated the internal sorption mechanisms using in situ FTIR, XPS, synchrotron XAS and DFT calculations. Compared with most previously reported gold sorbents, the developed pyrocarbon exhibited transcendental gold recovery performance, for its fordable production costs, satisfactory capacity, and easily separated properties. The Au(III) adsorption-reduction- nucleation processes were elucidated, in which the aromatic region in carbon matrices was proved play crucial role in forming reductive phenolic structures, and water molecules were found to participate in sorption processes; this is interesting and can enlighten us to pay more attention in chemical reactions in sorption. Through the specific material flow and cost calculations, the techno-economic analyses were employed to verify the cost-effectiveness of carbon-based gold sorption process. This paper offers a new and economically viable solution for gold recycling from aqueous solution. The experiments are reasonably well designed, and the mechanism interpretation are mostly supported by the data. In structural view, there are some points that need to be taken into consideration. Overall, it thinks this work is suitable for publication after minor revision considering following comments.*

Response: We appreciated for the reviewer for the highly positive comments on our work. We have carefully considered each point you raised and have conducted additional work to address these concerns. Please see the responses for the specific comments in the following texts.

Comment 1: *Pore diffusion plays a crucial role in sorption processes, particularly for the sorption of metal ions on char granules investigated in this study. BET analyses revealed an increase in specific surface areas for pyrocarbons and provided information on mesopores; however, micro pores with diameters less than 2 nm were not quantified (Supplementary Fig. 3b). It is imperative to provide more comprehensive details regarding the internal pore structure encompassing a wider range of pore sizes.*

Response: We sincerely appreciate the reviewer for the feedbacks on the properties of pore structures of the pyrocarbon sorbents. By following the reviewer's comments, the N₂ adsorption-desorption isotherm curves of the pyrocarbon materials were remeasured, in which we have optimized the methods by adding the measurement points within the low relative pressure regions ($P/P_0 < 0.02$), so as to acquire more pore information in micro pore regions (< 2 nm). As shown in Supplementary Fig. 3, the remeasured specific surface areas of our PyC500, PyC600, PyC700, and PyC800 were 565.7, 573.2, 597.6, and 673.8 m² g⁻¹, with their pore diameters centered at 3.79, 3.82,

3.96, and 3.97 nm, respectively. These results well coincided with the previous BET results on the pore structures of these sorbents (Fig. R1).

Updated Supplementary Fig. 3:

Supplementary Fig. 3. (a) N₂ adsorption-desorption isotherms and (b) pore size distribution of PyC500, PyC600, PyC700, and PyC800.

The previous submitted BET results:

Fig. R1. The previously measured (a) N₂ adsorption-desorption isotherms and (b) pore size distribution of PyC500, PyC600, PyC700, and PyC800.

We have updated the Supplementary Fig. 3 of BET results in Supplementary Information, and revised the relevant descriptions in the main text of manuscript as follows:

Revised description of BET analysis:

Line 137, “Notably, higher pyrolysis temperatures yielded increased SSAs for the PyCs (ranging from 565.7 to 673.8 m² g⁻¹) while maintaining similar mesoporous structures with pore diameters of 3.79 to 3.97 nm.”

Comment 2: *Except for the abbreviations used for characterizations (e.g., XPS, HRTEM), there are numerous other abbreviations in the manuscript, such as BCTs, BC500-800, PTs, Ph-OH, etc. It is essential to ensure that these abbreviations are necessary and do not confuse the readers.*

Response: We greatly appreciate your constructive feedback regarding our writing. We have thoroughly reviewed the entire manuscript and carefully revised the abbreviations used to enhance reader comprehension. In accordance with the reviewer's suggestions, we have eliminated abbreviations such as “PTs”, “Ph-OH”, and “PFRs” in order to provide clearer description. The abbreviations of “BCTs” and its related BC500-800 have been revised into forms of “PyCs” and PyC500-800 (e.g., PyC500, PyC600, PyC700, and PyC800). To avoid the confusion to the readers, we have also clarified this in the main text as follows:

Line 129: “Pyrolysis temperatures were purposefully adjusted to modulate the characteristics of the resulting PyCs (“PyCs” represents the as-synthesized pyrocarbon sorbent pyrolyzed at the temperatures of 500, 600, 700, and 800°C).

Comment 3: *The authors employed the term "pyrocarbon" to describe the prepared sorbent, while using the abbreviation "BC" for brevity. It is speculated that "BC" likely refers to biochar. Please ensure consistency in abbreviations with their original meanings.*

Response: We would like to express our gratitude to the reviewer for bringing to our attention the inappropriate use of abbreviations in the manuscript. In response to this valuable comment, we have thoroughly revised and clarified all ambiguous abbreviations throughout our manuscript description. “PyC” will be used as the abbreviation of the pyrocarbon sorbent, and those sorbents subjected to various temperatures during their respective pyrolysis processes were labeled as “PyC500”, “PyC600”, “PyC700” and “PyC800”, respectively. We express our sincere gratitude to the reviewer for their meticulous review of our manuscript.

Comment 4: *There are some written or illustrated details that require revision.:* (1) *Supplementary Fig. 23 should provide the full name of GRP for clarity;* (2) *In MS, it was stated that the purity of recycled gold from CPUs was 23.96 karat; however, this contrasts with the value shown in Fig. 2g (23.9 karat);* (3) *Line 859: Please correct "Nature Sustainability" to "Nature Communications". Please thoroughly review these points and meticulously examine the entire manuscript to prevent the occurrence of such errors.*

Response: We are sorry for the confusion. The suggested revisions have been made to address the points of confusion raised by the reviewer. Specifically, (1) The abbreviation “GRP” has been replaced with “gold recovery processes”; (2) The purity of recycled gold has been clarified as 23.96 karat and this correction is reflected in Fig. 2g; (3) The journal name has been corrected to “Nature Communications” for submission purposes. In order to enhance the quality of the article, we have thoroughly reviewed the entire manuscript to prevent such errors from occurring again. We sincerely appreciate the reviewer's careful reading and valuable suggestions.

Corrected sentences about the journal name:

Line 903: “Peer review information Nature Communications thanks ### and the other, anonymous, reviewer(s) for their contribution to the peer review of this work.”

Revised Supplementary Fig. 23:

Supplementary Fig. 23. Schematic diagram of structure evolution of PyC700 during the gold recovery process.

Revised Fig. 2c and 2d:

Fig. 2 Gold recovery performances. (a) The calculated Q_m and k_f of pyrocarbon from Langmuir model and mass transfer model. (b) adsorption kinetics of Au(III) using PyC700. (c) The comparison of Au(III) recovery capacity of PyC700 with that of previously published sorbents. (d) Effects of solution acid-alkaline conditions on Au(III) recovery by PyC700. (e) Effects of potentially interfering species including cations and anions on Au(III) recovery by PyC700. (f) The calculated K_a and the comparison of redox potentials for various metal ions. (g) Schematic of the gold recovery from CPU leachate using PyC700. The concentration change of Au(III) using PyC700 as sorbent in (h) AMD leachate and (i) Intel leachate. (j) The Au(III) recovery by PyC700 in AMD and Intel leachate systems. The abbreviations “Sorp.” and “Sorp. & Red.” in Fig. 2c and 2d signify that the reported sorbents functionalized through simple sorption and sorption-coupled reduction processes, respectively.

Comment 5: *In this work, the authors employed mass transfer coefficient k_f to distinguish the adsorption kinetics of those BCTs sorbents, however it suggests kinetic constants were used more frequently in fitting kinetics data. What are the advantages of using k_f compared to typical kinetic constants?*

Response: Thanks for your feedbacks regarding the kinetics models. The adsorption process is mass transfer limited and involves both the external mass transfer resistance and internal diffusion resistance. The external mass transfer resistance is characterized by a film diffusion coefficient, while the internal diffusion is either a part of pore diffusion or surface diffusion through pore walls. Our study employed a mass transfer model to quantitatively assess the adsorption rates of various sorbents. By examining the values of the mass transfer coefficients (k_f), we gained insights into the speed of the adsorption process, determining whether it is rapid or slow. As demonstrated in Supplementary Note 3, calculating k_f values involved an essential normalization procedure that considered inherent properties of sorbents (*e.g.*, specific surface area) and reaction conditions (*e.g.*, sorbent dosages). However, commonly used adsorption kinetics models, including the pseudo-first and second-order models, often overlook crucial factors such as the pore structure properties of sorbents and the diffusion steps of adsorbates, which play a vital role in most adsorption processes. Furthermore, these models do not explain mass transfer phenomena adequately and are not suitable for engineering designs. It should be noted that previous studies have widely employed the mass transfer model due to its ability to provide detailed information on adsorbate diffusion and transfer while accurately reflecting the mass transfer processes occurring within porous structures.¹⁻³ Based on these considerations, we adopted the mass transfer model in this study to quantitatively compare the kinetic performance of various pyrocarbon sorbents for gold recovery.

To provide clarification on the advancements and the purpose of utilizing the mass transfer model, we have supplemented the descriptions as follows:

Updated text in Supplementary Note 3: “*In this model, calculating k_f values incorporated a critical normalization procedure that considered various reaction details (*e.g.*, specific surface area, sorbent dosages) during sorption processes. It was widely employed in previous studies because it can effectively and adequately explain mass transfer phenomena and offers valuable insights into adsorbate diffusion and transfer. Furthermore, it accurately represents the mass transfer occurring within porous structures.²⁻⁴”*

Comment 6: *The OCPT depicted in Supplementary Fig. 16a suggests a recovery selectivity that is dependent on the reduction potential. This observation indicates the occurrence of electron transfer*

for Fe(III). What are the reduction products of Fe(III) ions? Could the presence of Fe(III) hinder the reduction of gold ions due to competition for electrons?

Response: These points raised by the reviewer are profound, we have indeed observed that ferric ions also attract electrons from the pyrocarbon electrode due to their relatively high redox potential ($E^0_{\text{Fe(III)/Fe(II)}} = +0.77 \text{ V}$). However, it is worth noting that our manuscript results indicate a lower Fe(III) adsorption efficiency (~42.5%) of our PyC700 sorbent (Supplementary Fig. 14b), which is five orders of magnitude higher than that of Fe(III) (Fig. 2f), demonstrating excellent selectivity towards gold ions over ferric ions. To address the reviewer's concerns regarding the reduction product of Fe(III), we further characterized the Fe(III)-sorbed PyC700 using XRD and XPS techniques. As shown in Supplementary Fig. 16b, no characteristic peaks assigned to Fe^0 appeared in the XRD pattern of PyC700 after reaction with FeCl_3 solution. Additionally, deconvoluted XPS Fe 2p spectra indicated the coexistence of both Fe(III) and Fe(II) on the Fe(III)-sorbed PyC700 (Supplementary Fig. 16c). These results convincingly confirm that there was partial reduction of Fe(III) to Fe(II) by PyC700, which aligns well with their respective redox potentials (E^0): Fe(III)/Fe(II) ($E^0_{\text{Fe(III)/Fe(II)}} = +0.77 \text{ V}$) and Fe(III)/ Fe^0 ($E^0_{\text{Fe(III)/Fe(0)}} = -0.037 \text{ V}$). Moreover, in Supplementary Fig. 14a, our results have verified the copresence of Fe(III) demonstrated no obvious interference on the gold recovery efficiency. Considering higher E^0 of Au(III)/ Au^0 ($E^0_{\text{Au(III)/Au(0)}} = +1.002 \text{ V}$) than Fe(III)/Fe(II) ($E^0_{\text{Fe(III)/Fe(II)}} = +0.77 \text{ V}$), it can be speculated that gold ions exhibited a significantly greater tendency to acquire electrons from PyC700 even when competing with ferric ions. These findings also support the assumption of reduction potential (E_0)-dependent selectivity of PyC700 towards different metal ions.

To clarify these findings, we have further added the characterization results as Supplementary Fig. 16b and 16c in Supplementary Information:

Supplementary Fig. 16. (a) The OCPT curves on the PyC700 electrodes in various competing cations solution systems. (b) XRD pattern of the PyC700 after the adsorption of Fe(III) in 500 mg L⁻¹ FeCl₃ solution. (c) XPS analysis of Fe 2p in a high-resolution of Fe(III)-sorbed PyC700 sample. (d) the K_d values for various coexisting cations with different reduction potential.

We also provided detailed discussion in the figure caption of Supplementary Fig. 16 with focus on the Fe(III) reduction product analysis as follows:

Figure caption of Supplementary Fig. 16: “Minimal electrons transfer implied the redox reaction occurring between Fe(III) and PyC700 electrode, as reflected by the OPCT fluctuation. Considering higher redox potential of Fe(III)/Fe(II) ($E^0_{\text{Fe(III)/Fe(II)}} = +0.77$ V) than Fe(III)/Fe⁰ ($E^0_{\text{Fe(III)/Fe(0)}} = -0.037$ V), the reductant can be most probably divalent Fe(II), rather than metallic Fe⁰. This speculation was well manifested by XRD and XPS results in Supplementary Fig. 16b and 16c. Specifically, no characteristic peaks assigned to Fe⁰ appeared in the XRD pattern of the PyC700 after reaction in FeCl₃ solution. And the deconvoluted XPS Fe 2p spectra indicated the copresence of Fe(III) and Fe(II) on the Fe(III)-sorbed PyC700. These results identified the partially reduction of Fe(III) to Fe(II) by PyC700; however, it exhibited no interference on the gold recovery performance, mainly

due to high redox potential of Au(III)/Au⁰ ($E^0_{\text{Au(III)/Au(0)}} = +1.002 \text{ V}$) and the sufficient reductive electron supply.”

Moreover, the relevant sentences in the main text of manuscript were also revised as follows:

Line 239: “In contrast, no electron transfer occurred in solutions of other cations, except for Fe(III). Potential fluctuation indicated minimal electron transfer between Fe(III) and PyC700 electrode, which was further unveiled by the XRD and XPS results that the Fe(III) was reduced to Fe(II) on the sorbent surface (Supplementary Fig. 16b, c). These findings suggest a reduction potential (E_0)-dependent selectivity of PyC700 towards different metal ions (Supplementary Fig. 16d).”

Comment 7: *The adsorption capacities for AuBr₄⁻ and Au(CN)₂⁻ exhibited a significant decrease, necessitating further discussion to elucidate the underlying reasons for this phenomenon. Additionally, it is imperative to clarify the adsorptive and/or reductive forms of these two Au species.*

Response: Thanks for the meaningful insights. As discussed in the manuscript, chemical reduction processes played a dominant role in the gold recovery procedure. The distinctive capacity fluctuations of these three Au-anion complexes can be speculated to be highly dependent on their reduction behaviors, which were influenced by both the redox potential (E_0) and coordination stability constants ($\lg K_f^0$).⁴ Specifically, the E^0 values for the three Au-anion complexes are as follows: AuCl₄⁻ (1.002 V) > AuBr₄⁻ (0.854 V) >> Au(CN)₂⁻ (-0.59 V), and thus the PyC700 exhibited varying capability to contribute its delocalized electrons toward these Au species. For instance, due to negative E^0 , Au(CN)₂⁻ is too difficult to acquire electrons to be reduced. In another aspect, decomplexation of ligands from Au-anions is also essential for the gold reduction and nucleation process, and it becomes more challenging with higher $\lg K_f^0$ values. The $\lg K_f^0$ values for the three Au-anion complexes are as follows: Au(CN)₂⁻ > AuBr₄⁻ > AuCl₄⁻. Higher $\lg K_f^0$ values of Au(CN)₂⁻ make it harder to detach ligands and form new bonds between Au atoms. In conclusion, variations in performance of PyC700 towards different anionic forms of gold can mainly be attributed to differences in their E^0 and $\lg K_f^0$ values. Considering the potential variation of leaching agents for specific purposes, it is crucial to acknowledge that all these complexes involving Au-anions represent significant forms of gold in diverse leaching solutions. Further efforts are required to develop advanced decomplexation technologies targeting stable Au-anions. These endeavors will contribute to the promotion of sorbent-based gold recycling technology and enhance environmental sustainability in wide range application in different water matrixes.

To address the reviewer's concerns on the recycled forms of Au on the sorbent, the XRD patterns of AuBr_4^- and $\text{Au}(\text{CN})_2^-$ were further obtained as provided in Supplementary Fig. 17e:

Supplementary Fig. 17. Adsorption isotherms for (a) AuBr_4^- and (b) $\text{Au}(\text{CN})_2^-$ adsorption onto PyC700, (c) OCPT curves on the PyC700 electrodes in various Au(III) species solution systems. (d) maximum recovery capacity for various gold complexes with different reduction potential (E_0). (e) XRD patterns of the PyC700 after the gold sorption in 500 mg L^{-1} AuCl_4^- , AuBr_4^- , and $\text{Au}(\text{CN})_2^-$ solution, respectively.

As shown in Supplementary Fig. 17e, the characteristic Bragg reflections of Au at 38.2° and 44.4° emerged in the XRD patterns of AuBr_4^- and AuCl_4^- -loaded PyC700, which were assigned to the (111) and (200) crystal planes of Au^0 . These observations implied that both AuCl_4^- and AuBr_4^- can be successfully sorbed onto PyC700 and consequently reduced into metallic Au^0 . However, these peaks were not found in $\text{Au}(\text{CN})_2^-$ -loaded PyC700, suggesting that $\text{Au}(\text{CN})_2^-$ can only be simply captured and enriched on the PyC700, maintaining its original form due to its low E^0 and high $\lg K_f^0$.

To clarify these underlying reasons and findings, we have revised the relevant description in the manuscript as follows:

Line 247: “Our findings revealed a moderate decrease in measured Q_m values by 66.9% for AuBr_4^- while a significant decrease of 95.7% was observed for $\text{Au}(\text{CN})_2^-$ (Supplementary Fig. 17a, b and Table 6). The hypothesis that adsorption behaviors depend on reduction potential effectively clarifies this phenomenon (Supplementary Fig. 17c, d). It suggests that transferring electrons to AuBr_4^- becomes increasingly difficult, and it's even less probable for $\text{Au}(\text{CN})_2^-$ to be reduced to metallic Au^0 (Supplementary Fig. 17e). This phenomenon can also be attributed to the differences in the coordination stability constants ($\lg K_f^0$) among these three complexes with Au-anions complexes ($\lg K_f^0(\text{Au}(\text{CN})_2^-) > \lg K_f^0(\text{AuBr}_4^-) > \lg K_f^0(\text{AuCl}_4^-)$), as higher $\lg K_f^0$ value indicates greater difficulty in decomposing the ligands from Au-anions to form metallic Au-Au bonds.⁵⁹”

Comment 8: *The cations of Al(III), Fe(III), Ni(II), Cu(II), Zn(II) were used to investigate the adsorption selectivity. It is essential to clarify the selection criteria for these cations in the MS. Moreover, it is crucial to determine whether these cations are present in the actual E-waste leachates prepared in this study.*

Response: We sincerely appreciate the reviewer for providing feedback on the selectivity studies. The electronic wastes (E-waste) typically contain metals such as Cu, Fe, Al, Ni, Zn, Au, *etc.*, with variations in their component fractions depending on the specific types of E-wastes. For instance, Ilyas et al. conducted a chemical analysis of E-waste scraps and found that the major metals present were Cu (8.5%), Fe (8.3%), and Zn (8.1%), along with significant amounts of Pb (3.2%) and Ni (2.0%).⁵ To determine representative levels of cations in real leachate samples, we also referred to several gold-sorption-related studies. Lin and coworkers developed polyethyleneimine-starch fibers for gold adsorption from actual E-waste leachate and identified Al(III), Fe(III), Ni(II), Cu(II), and Zn(II) as typical interfering cations used to study sorbent selectivity.⁶ Shu and coworkers recently reported a novel porphyrin-based MOF sorbent for gold recovery, which also highlighted Al(III), Fe(III), Ni(II), Cu(II), and Zn(II) as prominent constituents in the leachate.⁷ According to a series of previous studies,^{8, 9} it has been consistently concluded that these five cations are universally recognized as the primary constituents in actual E-waste leachate. Therefore, these representative cations were identified as significant coexisting interfering species in actual E-waste leachate and were selected to assess the selectivity of our sorbent for gold recovery in subsequent studies.

To clarify these points, we have discussed the selection criteria of those cations as illustrated in the **Method** section:

Line 555, “Based on the typical components in E-wastes,^{16, 72} Al(III), Fe(III), Ni(II), Cu(II), and Zn(II) were selected as representative competing cations in the selectivity studies.”.

By following the reviewer's suggestion, cationic concentrations in the four types of CPU leachates were further determined using an inductively coupled plasma mass spectrometer. The measurement results have been provided as Supplementary Table 7 in the Supplementary Information:

Updated Supplemented Table 7:

Supplementary Table 7. Concentrations of typical cations in four types of leaching solutions of AMD and Intel CPUs.

Cations	AMD-Aqua	AMD-NBS	Intel-Aqua	Intel-NBS
Au(III)	1330	650.3	12.65	2.633
Na(I)	7.332	5.265	62.57	116.1
K(I)	1.136	1.577	1.949	9.591
Ca(II)	17.917	6.584	21.66	15.63
Mg(II)	2.616	1.576	2.852	3.333
Fe(III)	10108	578.6	102.6	39.67
Ni(II)	7035	157.6	2211	325.6
Cu(II)	29.55	22.5	33884	19839
Al(III)	35.24	28.79	1.919	1.157
Zn(II)	21.26	25.65	0.776	1.214
Mn(II)	1.072	1.781	1.525	2.129

Comment 9: *For the techno-economic analyses of sorbent production, more details regarding the specific material flow should be provided, such as material ratios, reaction parameters, etc. Additionally, it is necessary to clarify price parameters (e.g., electricity costs, manufacturing device expenses) that may impact the overall costs.*

Response: We express our gratitude to the reviewer for this constructive suggestion. By extensively referring to previous literature and published guidelines, it has been demonstrated that the production process of PyC700 is suitable for industrial use, taking into account various required

characteristics such as water tolerance, rapid mass transfer, low cost, and minimal environmental and safety impact. Supplementary Note 7 provides detailed information used in technoeconomic analysis (TEA), including selection criteria for electricity prices, workers' salaries, CPU scrap prices *etc.* Specifically, Supplementary Note 7 provides price parameters (*e.g.*, electricity costs and manufacturing device expenses) used in calculating total costs which we have extracted below:

Selection criteria of electricity price: “Of note, the electricity price selected in this work (0.02 ~ 0.06 \$ kWh⁻¹) was based on the recently published literature in which TEA were carried out for electrosynthesis of ethylene.”.

Calculation of electricity cost: “(2) Operating costs. The operating costs were considered to include the electricity, maintenance, labor, depreciation, and other operating costs (*e.g.*, administrative cost, insurance fee, *etc.*). The selected electricity price (0.06 \$ kWh⁻¹) was based on local electricity price in Shanghai, China, a combination of peak- and valley-time prices.”;

“Electricity: = (0.3 kW×2 h day⁻¹+0.02 kW×6 h day⁻¹+0.2 kW×6 h day⁻¹+1.6 kW×24 h day⁻¹+1.2 kW×6 h day⁻¹+0.38 kW×12 h day⁻¹)×365 day×0.06 \$ kWh⁻¹=\$1140.55”;

Manufacturing device expense: “Capital cost. The total capital cost was considered to include the equipment purchase cost and other capital costs. To be noted, the other capital costs were typically generated due to equipment installation, process piping, instrumentation and controls, electrical systems, *etc.* These costs are difficult to determine as they depend on many factors, and some approximations exist to estimate these costs as a percentage. To ensure the accuracy of the results, we have referred to the methods previously reported to calculate the capital costs for producing honeydew peel activated carbon.¹²

1. Equipment purchase cost:

= \$27.43 (laboratory water bath)+\$274.35 (peristaltic pump)+\$20.58 (Magnetic stirrer)+\$685.87 (freeze dryer)+\$1646.09 (tube furnace)+\$548.70 (thermostatic oscillator)+\$41.2 (glass wares)=\$3244.17”

The costs used for purchasing manufacturing devices have also been provided in Supplementary Fig. 49 as follows:

Supplementary Fig. 49. (a) Material flow analysis of the specific example of gold recovery process from CPU scrap using PyC700. The specific information of (b) the equipment and (c) the chemicals/materials involved in TEA for gold recovery based on PyC700.

As for the details of the specific mass flow in TEA, such as material ratio and reaction parameters, we have additionally provided a detailed description in the subsection of Supplementary Note 7:

Description of TEA Details: “The material ratios, reaction parameters (working temperature, reaction time, *etc.*), and the yield of the products employed during TEA were determined with reference to laboratory scale realities. We have summarized the technological parameters of the electric equipment during the gold recovery processes in Supplementary Fig. 49.”;

and “In accordance with the actual experimental conditions during our lab-scale production of PyC700, the calculation details such as material ratios and reaction parameters were carefully selected. Specifically, we assumed a daily output of 20 g day⁻¹ for the sorbents, resulting in an annual production of 7.3 kilograms for PyC700. The annual running times for the equipment were determined based on their actual working time in preparation processes, which are provided in Supplementary Fig. 49.”;

and “The annual usage of these chemicals was calculated based on their daily consumption in the lab-scale production processes, which are provided in Supplementary Fig. 49.”

To better clarify the TEA methodologies and calculation details in the **Method** section in main text of manuscript, we also revised the relevant sentence as follows:

Line626: “Details of the TEA methodologies, and the specific sample for the calculation of total cost of the proposed gold recovery process from CPU scrap using pyrolytic PyC700 could be found in Supplementary Note 7.”.

We would like to express our sincere gratitude again to the reviewer for their invaluable assistance in enhancing the logical coherence and rigor of our TEA analysis. Only by establishing a comprehensive and reliable foundational database in TEA can we furnish precise and compelling economic outcomes, thereby facilitating an accurate evaluation of the cost-effectiveness and economic viability of our pyrocarbon sorbent-based gold recovery strategies.

Reviewer #2

General comment: *The work describes the design and mechanistic study of pyrocarbons for the recovery of gold. The study describes the nucleation of gold on the functional groups at the surface of the carbon, and a combined DFT study to explain the mechanism. Finally, a TEA and proof-of-concept for real-world CPU is used. While the work is certainly of technical interest, and the redox-transformation of gold brings significant benefits, the overall novelty of the work is not considered sufficient for Nature Communications.*

Response: We extend our deepest gratitude to the reviewer for their detailed and thorough feedback, which plays an essential role in maintaining the high standards of articles published in Nature Communications. Our research introduces a groundbreaking green carbon-based sorption technology through the nuanced tailoring of hybridization levels in pyrocarbon. This innovation facilitates the highly selective and efficient extraction of gold from E-waste leachate. The methodological ingenuity in sorbent development, the unique mechanisms underlying gold recovery, and the extensive, meticulous evaluation of its applicability to actual E-waste leachate processing underscore the novelty of our study. We are confident that our work not only embodies significant innovation but also aligns seamlessly with the editorial focus of Nature Communications. We fully acknowledge the reviewer's concerns and wish to assure them that the issues they have highlighted, especially those pertaining to the novelty of our research and the accuracy of the sorption capacity comparisons, will be meticulously and thoroughly addressed in our revised manuscript. We are committed to conducting a comprehensive and rigorous examination to resolve these concerns, thereby enhancing the clarity and impact of our submission. Our dedicated to ensuring that our revised work meets the high standards expected by the journal and reviewer to contributes meaningfully to the field. We appreciate the opportunity to delve into specifics and have elaborated on our responses below.

Special Comments:

Comment 1: *Activated carbon and other forms of carbons that contain aromaticity and phenolic groups have been extensively used as traditional methods for gold adsorption, including at an industrial scale, and while there are interesting mechanistic insights on the particular carbon chemistry shown in this work, the framework for separation remains known. Therefore, the work is of more specialized and incremental technical interest than transformative. Therefore, a more specialized journal in sustainability or separation technology is recommended for the work.*

Response: Thank you for your feedback. Our study introduces an innovative methodology aimed at extracting gold from E-waste leachate using an economical pyrocarbon sorbent derived from abundant and sustainable sources like seaweed-based alginate. By carefully adjusting factors like the balance between graphitization and amorphism, electrical conductivity, structural imperfections in the carbon, and functional groups on the surface, we have improved the efficiency of the pyrocarbon sorbent in recovering gold. These tailored properties are key to our strategy for achieving exceptional efficiency and specificity in the gold extraction process, and reducing both environmental footprint and operational expenses. Moreover, we have carefully detailed the key findings of our research to underscore the originality of our approach, as outlined in the following sections:

(1) Novelty on the pyrocarbon sorbent:

The recent surge in developing innovative materials for gold sorption and reduction showcases significant advancements in the field. These include molybdenic sulfide, metal-organic frameworks (MOFs), covalent organic frameworks (COFs), and porphyrin polymers. Efforts have been made to tailor various properties of these materials in order to enhance their performance. Specifically, the porosity, sorption affinity, electron contribution, and transferring capability are considered as crucial aspects that need to be customized for optimal gold sorbents. Yet, an important consideration is the cost and environmental footprint of producing these advanced sorbents. Despite their promising capabilities, the synthesis of many such materials involves procedures that are either financially burdensome or detrimental to the environment. This dichotomy underscores the necessity for ongoing research to not only improve the efficacy of these sorbents but also to devise methods for their sustainable and eco-friendly production, ensuring that the pursuit of technological advancement does not come at the expense of environmental well-being. In contrast to the aforementioned materials, our proposed pyrocarbon offers a compelling alternative with its cost-effectiveness and promising attributes for practical gold sorption applications. Its synthesis from calcium alginate beads through pyrolysis followed by acidic rinsing is not only straightforward but also environmentally benign, yielding pyrocarbon granules that are readily separable post-use. This process also effectively removes inorganic CaO particles embedded in the carbon matrix, yielding a highly porous structure that enhances mass transfer coefficient (k_f of $1.73 \times 10^{-8} \text{ m s}^{-1}$). These attributes not only underscore the practicality and environmental friendliness of our pyrocarbon but also highlight its potential as an efficient, sustainable option for gold recovery in contrast to its more resource-intensive or environmentally challenging counterparts. These advantages significantly surpass those of other reported gold sorbents. While the separation of reduced Au⁰ nanoparticles from their powdered counterparts often presents substantial hurdles, our method involving

calcination and acidic rinsing efficiently yields high-purity (23.96 karat) gold directly from the Au-sorbed pyrocarbon. This process is not only user-friendly but also economically advantageous, offering a streamlined and cost-effective route to gold recovery.

Furthermore, while a considerable body of research has focused on carbon-based gold sorbents (e.g., activated carbon,^{10, 11} carbon nanotubes,^{12, 13} biochar,^{14, 15} and graphene oxide¹⁶⁻¹⁸), these studies predominantly target the enhancement of mesoporous structures or the addition of functional groups conducive to sorption (e.g., phenolic hydroxyl,¹⁷ sulfhydryl groups). In contrast, our pyrocarbon sorbent stands out for its unique synthesis and post-treatment processes, which not only improve its structural properties for effective gold uptake but also facilitate the subsequent recovery of pure gold, establishing a novel benchmark in the field of gold sorption technology. By following this, we have revised the Supplementary Table 4 and summarized numerous previously carbon-based gold sorbents in this table. Notably, the sorption capacity (Q_m) of those sorbents were mostly lower than 500 mg g⁻¹, except that of reduced graphene oxide and phenolic hydroxyl-rich graphene oxide.^{17, 19} The interplay between structure and function in carbon-based sorbents, particularly concerning the combined influence of graphitic domains and reductive functional groups like phenolic hydroxyl, remains inadequately explored and understood. This gap highlights the necessity of examining how these structural elements synergistically enhance gold sorption capabilities. In our research, we have intentionally engineered the degree of pyrocarbon graphitization and optimized its surface functionalities, alongside other vital structural attributes, to promote effective gold ion capture while reducing unwanted interactions with other metallic ions and impurities. As a result, our pyrocarbon not only demonstrates superior Q_m but also exhibits enhanced sorption kinetics when compared to traditional carbon-based sorbents. These advancements signify a major leap forward, positioning our pyrocarbon as a groundbreaking alternative in the realm of gold recovery technologies. The superior performance and the sustainable, cost-effective production process of our pyrocarbon redefine its utility and effectiveness, distinguishing it as a significant advancement over existing materials in the field of gold recycling.

(2) Understanding the structure-function relationship in pyrocarbon:

As highlighted earlier, the limited understanding of structure-function relationship of pyrocarbon has constrained its efficacy and practical deployment in gold recovery operations. The absence of targeted studies to elucidate the manipulation of pyrocarbon properties for optimized gold recycling further underscores this gap in the field. Our research is designed to bridge this knowledge chasm by pinpointing and investigating the key attributes of pyrocarbon. We meticulously examine its interaction dynamics and the structural and functional transformations it undergoes during the gold recovery process, aiming to unlock and enhance its sorption potential.

Through comprehensive analyses, encompassing both advanced characterization techniques and theoretical calculations, we have meticulously deciphered the underlying mechanisms governing electron donation and transfer within pyrocarbon. Our investigation delves deep into the intricate processes that facilitate these electron interactions, shedding light on how pyrocarbon's unique structure contributes to its efficacy in gold recovery. This enhanced understanding not only clarifies the role of electron dynamics in pyrocarbon's function but also provides a foundational basis for optimizing its design and application in gold sorption technologies. Specifically, our research has pinpointed that aromatic structures within graphitic domains contribute π conjugated delocalized electrons through hydroxylation processes, wherein intrinsic and adjacent phenolic hydroxyl groups act as pivotal "switches" facilitating the electron transfer to gold ions, thereby inducing their reduction. This discovery reveals that both graphitic and oxygen-containing amorphous structures are instrumental in the gold reduction process, underscoring their critical roles. Such insights gleaned from our study contest the conventional belief that attributes such as porosity or the presence of functional groups with high sorption affinity are the primary determinants of gold recovery efficiency. Instead, we highlight a nuanced mechanism where electron donation and transfer dynamics within specific structural frameworks of pyrocarbon are key to enhancing gold recovery.

Here's a detailed breakdown of these aspects:

Degree of Graphitization: The extent to which carbon atoms in pyrocarbon are ordered (graphitized) can significantly influence its electrical conductivity and chemical stability. Higher graphitization levels can enhance the material's ability to contribute and transfer electrons, potentially facilitating the reduction of gold ions to elemental gold. This aspect is particularly important for the reductive deposition of gold from solutions.

Porosity: The porosity of pyrocarbon, including both micro and mesopores, directly impacts its surface area, which in turn affects its mass transfer and also sorption capacity. A highly porous structure provides more active sites for gold ion adsorption and facilitates easier access to these sites, improving the sorption efficiency.

Surface Area: The increased surface area facilitates the availability of binding sites for ion capture, thereby enhancing the interaction between pyrocarbon and gold ions and promoting intensive reduction processes, ultimately resulting in a higher recovery capacity. Tailoring the surface area through the pyrocarbon's synthesis and processing conditions is key to enhancing its performance as a sorbent.

Functional Groups: The presence and nature of functional groups on the surface of pyrocarbon play a pivotal role in its sorption behavior. Groups such as phenolic hydroxyls can interact with gold ions, initiating the start-up stage of the stabilization and reduction of gold ions on the sorbent surface. The careful introduction and modulation of these groups can improve selectivity and affinity for gold over other metal ions.

Interaction with Gold Ions: The effectiveness of pyrocarbon in gold recovery hinges on its interactions with gold ions. These interactions are influenced by the aforementioned structural features, which dictate how well the pyrocarbon can capture, reduce, and hold onto gold ions.

In sum, the structure-function relationship in pyrocarbon is a multifaceted concept that requires a comprehensive understanding of how various structural attributes influence its role as a gold sorbent. By elucidating this relationship, researchers can tailor pyrocarbon's properties to maximize its sorption performance and efficiency in gold recovery applications.

(3) Proposing "Stepwise" polymerization and nucleation reduction mechanism:

Beyond the distinctive attributes of pyrocarbon, our research also introduces an innovative "stepwise" nucleation reduction mechanism that enriches the fundamental understanding in gold reduction process for its recovery from E-waste leachate. Building on fundamental concepts of gold chemistry and surface interactions, our mechanistic model provides a detailed account of the complex processes that facilitate the reduction of gold ions and encourage their nucleation on the pyrocarbon sorbent surface. This mechanism unfolds in distinct, sequential stages, beginning with the initial interaction between gold ions and the pyrocarbon surface, followed by a series of intricate transformations (*e.g.*, Adsorption-Polymerization-Nucleation-Growth-Aggregation to Metallic Gold) that culminate in gold reduction and recovery. Previously, classical nucleation theory posited that metallic ions could transition directly from their higher valent states to zerovalent atoms, subsequently aggregating to form metal nuclei. These nuclei would then expand via processes such as Ostwald ripening or coalescence. This theory has traditionally been applied to elucidate the adsorption mechanisms observed in gold adsorption studies, offering insights into the initial stages of metal ion reduction and the subsequent growth patterns of metallic particles on adsorbent surfaces. For instance, previous studies have clarified that AuCl_4^- species undergo dichlorination reduction during gold recovery by releasing four chlorine ions to generate zerovalent gold atoms, which subsequently agglomerate into clusters (*e.g.*, Au_3 , Au_4) on the surfaces of adsorbents.^{9, 20} In contrast, our research reveals a different initial pathway: the reduction of gold ions starts with the formation of chlorine-protected Au-Cl clusters on the sorbent surface, rather than the immediate generation of individual zero-valent gold atoms. Over time, these clusters undergo a "stepwise" reduction, wherein chlorine ions are methodically detached, significantly accelerating the gold reduction

process by lowering the energy barriers from 1.08 to -21.84 eV. This nuanced reduction mechanism is particularly relevant for gold ions, which are known to form highly stable complexes with chlorine ions ($\lg K_f^p=17.40$), distinguishing the behavior of gold from other metals like silver (Ag(I)). This stepwise reduction underscores a unique aspect of gold ion chemistry and provides a more energy-favorable pathway for gold recovery. The detachment of all chlorine ions directly from AuCl_4^- species may encounter high energy barriers; instead, unreduced Au^{x+} ($x=0-3$) species could form intermediate gold-chlorine complexes, thereby reducing the overall reduction barriers.²¹ Our proposed mechanism is distinguished by its "stepwise" character, featuring clearly defined intermediate stages, each with its own specific energy barriers and activation energies. This structure allows for the efficient harnessing of contributed electrons and minimizes energy dissipation throughout the process. To substantiate the credibility of our proposed mechanisms, we employed a comprehensive suite of analytical techniques. Transmission electron microscopy (TEM) provided direct visualization of the nucleation and growth of gold particles. X-ray absorption fine structure (XAFS) spectra offered insights into the electronic structure and coordination environment changes during the reduction process. Lastly, density functional theory (DFT) calculations were instrumental in quantifying the energy profiles and activation energies associated with each step, providing a robust theoretical framework that aligns with our empirical observations. Collectively, these analyses not only validate our stepwise mechanism but also illustrate its significance in advancing our understanding of gold recovery processes. Our thorough investigations deliver convincing proof of intermediate gold-chlorine clusters and illuminate the thermodynamic and kinetic principles steering nucleation and growth phenomena on pyrocarbon substrates. Additionally, the mechanistic framework we propose provides critical insights into the determinants of efficiency and selectivity in gold recovery operations, enabling targeted optimization approaches for subsequent research in this domain. By elucidating the complex dynamics of gold ion reduction on pyrocarbon sorbents, our research enhances the foundational knowledge of metal sorption mechanics. This contribution is pivotal in paving the way for the creation of more effective and environmentally sustainable metal recovery methodologies, marking a significant step forward in the field of material science, metal recycling and sustainable environmental remediation.

(4) Economic viability and environmental sustainability:

Economic viability and environmental sustainability are two critical pillars that must be integrated into the development and implementation of any new technology, especially in the context of materials science and metal recovery. Our study not only heralds technical innovations but also vividly highlights the economic viability and environmental sustainability of our advanced gold recovery methodology. Through comprehensive technoeconomic analysis, we have ascertained

that our pyrocarbon-based sorption strategy yields extraordinarily advantageous input-output economic ratios, achieving potential cost savings exceeding 1370%. This significant economic benefit, in conjunction with our reliance on eco-friendly materials and processes, clearly positions our approach as a superior and innovative alternative to the conventional methodologies that typically require expensive reagents and high-energy inputs. Moreover, our approach eschews the utilization of organic solvents and noxious chemicals that are staples in traditional gold recovery techniques, markedly diminishing environmental pollution and reducing health risks. By embracing and integrating the principles of the circular economy and green chemistry, our research not only propels the electronics recycling industry forward but also resonates with the wider objectives of sustainable resource management and environmental preservation. Consequently, our work lays down a solid foundation for future endeavors aimed at boosting the sustainability and efficiency of metal recovery methodologies, contributing significantly to the progression towards more ethical and sustainable industrial practices.

In summarizing, our investigation has significantly broadened the horizons of carbon-based sorption theories, delving into the realm of carbon hybridization and spotlighting the pivotal roles that graphitic and oxygen-rich amorphous structures play in the extraction of gold from E-waste leachate. This breakthrough was facilitated by the strategic deployment of an economically viable pyrocarbon sorbent, coupled with the detailed characterization of an innovative "stepwise" nucleation reduction mechanism. Our research not only enriches the theoretical landscape of metal sorption but also furnishes actionable insights, potentially revolutionizing the efficiency and environmental sustainability of gold recovery methodologies. Thus, our study represents a significant leap forward, offering new perspectives and tools for advancing the domain of resource recovery. Through an integrated approach that combines rigorous experimental research, sophisticated theoretical modeling, and comprehensive technoeconomic analysis, we have established the efficacy, specificity, and eco-friendliness of our gold recovery method.

In addressing the novelty aspect of our work, we have highlighted its distinct contributions and scientific value, emphasizing that our findings represent not just incremental progress but substantial, innovative insights that address critical board aspects, such as environment, resources recovery and economic issues. Our confidence is high that this study will catalyze further investigation and innovation in the domain of sustainable metal recovery, aligning with the principles of a circular economy in the electronics sector. We are open to and encourage constructive critique to further corroborate and refine our results, aiming to foster the widespread adoption of environmentally responsible technologies and methodologies in a wide range.

To better clarify the above aspects, we have revised the abstract and other relevant descriptions as follows:

Line 34, “The sorbent demonstrates superior gold recovery performance compared to most previously reported advanced sorbents, showcasing its high recovery capacity of 2829.7 mg g⁻¹, high efficiency (>99.5%), remarkable selectivity ($K_d \sim 3.1 \times 10^8$ mL g⁻¹), and robust anti-interference capabilities within environmentally relevant contexts.”;

Line 84, “Efforts have been made to extract gold ions using various pyrocarbon-based materials, such as activated carbon,^{25, 26} carbon nanotubes,^{27, 28} biochar,^{29, 30} and graphene oxide³¹⁻³³, with predominant focus on the regulation of mesoporous structures or functional groups conducive to sorption (*e.g.*, phenolic hydroxyl,³² sulfhydryl groups³⁴). However, the limited understanding of the structure-function relationship of these pyrocarbon sorbents has constrained their efficacy and practical deployment in gold recovery operations.”;

To emphasize the technical advancement of our pyrocarbon, and also help compare its performance with other previous carbon-based sorbents, we also revised relevant sentences and Supplementary Table 4 as follows:

Line204, “This exceptional capacity surpasses other carbon-based sorbents (*e.g.*, activated carbon, carbon nanotube, graphene) and advanced nano sorbents (*e.g.*, MOFs, COFs, and metal sulfides), regardless of whether they rely on single sorption force or combined sorption-coupled reduction mechanisms (Supplementary Table 4).”

Updated Supplementary Table 4:

Supplementary Table 4. Comparison of Au(III) recovery performance of previously reported adsorbents.

Gold adsorbent	T (°C)	pH	Q_m (mg g^{-1})	K_d (mL g^{-1})	Recovery mechanism	Reference
NH ₂ -UiO-66-BA	-	2.5	1040	-	S&R ^[a]	13
COF-HNU25	25	7	1725	6.9×10^6	S&R	14
JNM-100-AO	25	2	954	2.9×10^6	S ^[b]	15
DTDD-MOF	25	2	1119	-	S&R	16
PCN-225	25	3	2613	4.8×10^7	S&R	17
GCC51	25	2	882	1.4×10^4	S&R	18
UiO-66-TA	25	2	374.8	2.4×10^5	S&R	19
CNT-MoS ₂ (2H)	-	4.8	2495	5.6×10^5	S&R	20
porphyrin polymer	-	2	1540	-	S&R	21
UiO-66-TU	25	4	326	-	S&R	22
UiO-66-ATU	25	3	227.7	-	S&R	23
UiO-66-BTU	25	2.5	680	1.0×10^4	S&R	24
Fe-BTC/PpDDA	28	2	934	1.3×10^4	S&R	25
RS-SR-NH-SiO ₂ -Fe ₃ O ₄	25	6	222	-	S	26
IM-TUCS	30	2	933	1.0×10^4	S	27
SH-MCM-41	20	2.5	195	-	S	28
UiO-66-NH ₂	25	2.5	650	-	S&R	29
CuS NPs	25	1	574.7	3.1×10^6	S&R	30
MoS ₂ NFs	25	1	1133	-	S&R	31
IECS-GLA	25	3	808.9	2.4×10^4	S	32
PGMA-NH ₂	25	4	1623	-	S&R	33
Aginate beads	30	1	387.7	-	S&R	34
PEI-modified sorbent	25	1	285	-	S&R	35
Thiol-ene hydrogel	25	0.5	118.8	-	S	36
Diatom biochar	25	2.9	443	4.7×10^3	S&R	37
Barley straw carbon	30	-	289.6	-	S&R	38
Activated carbon fiber	-	-	49.3	-	S&R	39
Activated carbon	-	3	32.1	-	S&R	40
Commercial activated carbon	-	3	33	-	S&R	40

Gold adsorbent	T (°C)	pH	Q_m (mg g ⁻¹)	K_d (mL g ⁻¹)	Recovery mechanism	Reference
rGO	25	4	1850	-	S&R	3
GO	25	6	108	-	S	41
PT-GO	30	3	1325	-	S	42
POPAM-grafted MWCNT	25	3	97	-	S	43
MWCNT	-	-	93.5	-	S&R	44
	10	2.75	1095.0	-	S&R	This work
BC700	25	2.75	2829.7	3.1×10^8	S&R	This work
	60	2.75	6368.4	-	S&R	This work

[a] The term “S&R” suggests that a sorption-coupled reduction mechanism governs the entire gold recovery process rather than simple sorption.

[b] The term “S” indicates that the gold recovery process relies solely on sorption processes without any valence change for enriched gold ions.

Comment 2: *A major technical comment is that a comparison between surface adsorption type mechanisms, as shown for many of the published data in 2d, and a gold reduction process, may be an unfair comparison as nucleation can give higher adsorption capacities. A more relevant comparison would be with other reduction methods (e.g., chemical reduction or electro-deposition), due to the possibility of multi-layer and higher overall areal capacity.*

Response: Thank you for the insights. The high redox potential of Au ($E^0_{\text{Au(III)/Au}} = +0.85$ to $+1.50$ V) indeed facilitates the reduction of gold ions in the presence of reductive functional groups like sulfhydryl, phenolic hydroxyl, and amine groups on the surface of sorbents. This process underscores the crucial role these functional groups play in the initial reduction stages, where they act as electron donors to reduce Au(III) to Au(0). Once the metallic Au⁰ nanoparticles form, their high surface free energy—referred to as a “boiling state”—significantly enhances their reactivity. This heightened reactivity is pivotal for electron transfer, promoting further reduction of gold ions in proximity to these nanoparticles.^{22, 23} Consequently, this leads to an autocatalytic effect where the

initial Au⁰ nanoparticles facilitate the reduction of additional gold ions, fostering the self-assembly and growth of gold aggregates. Notably, most sorbent reported previously were elaboratively designed, and characterized with inherent or decorated reductive moieties, such as amine groups in Fe-BTC/PpDDA²⁴ and NH₂-UiO-66-BA,⁹ and unsaturated sulfur species in MoS₂ nanoflakes.²⁵ This elucidation indeed clarifies why gold reduction predominantly transpires on the surfaces of such sorbents. Furthermore, it is crucial to recognize that the mechanisms of chemical reduction induced by reductive agents and electro-deposition, as mentioned by the reviewer, are inherently distinct from the sorption-based processes we are investigating. Chemical reduction occurs homogeneously within the solution, where reductive agents directly interact with gold ions, facilitating their reduction without the intermediary role of a solid sorbent. On the other hand, electro-deposition involves the transfer of electrons from a cathode within an electrochemical cell, where gold ions are reduced and deposited directly onto a conductive surface. These methods are fundamentally different from the sorption and electron donation/transfer phenomena observed at the solid-liquid interfaces between sorbents and gold-containing solutions. Our focus is on elucidating the interactions at these interfaces, where the sorbent not only captures gold ions but also facilitates their reduction through surface-bound functional groups. This distinction is crucial for understanding the unique contributions of our study, which aims to explore and optimize these interface-specific reactions for enhanced gold recovery. By clarifying these methodological differences, we emphasize the novelty and relevance of our research in the context of sorbent-based gold recovery technologies.

In our study, the patterns observed in published data, particularly in Figures 2c and 2d, resonate with similar mechanism (*e.g.*, sorption-reduction), albeit tailored to specific sorbents. Hence, drawing comparisons between the sorption performance of our pyrocarbon and these various sorbents is both logical and justifiable. Furthermore, the chemical reduction using reductive agents and electro-deposition suggested by the reviewer are predicated on fundamentally distinct mechanisms. Specifically, the former process involves gold ions reacting homogeneously within a solution, while the latter relies on acquiring electrons from a cathode within an electrochemical apparatus. Both methods diverge significantly from the exploration of sorption behaviors at the solid-liquid interfaces between sorbents and gold solutions. Consequently, we maintain that the mode of performance comparison employed in our study is well-founded and see no necessity for substantial revisions in this regard. We extend our sincere thanks to the reviewer for insightful and constructive comments concerning the logical coherence and precision of our performance comparison analysis. To address the inquiries regarding whether the sorption capacities observed were a result of singular sorption mechanisms or coupled sorption-reduction processes, we have undertaken the following revisions to Figures 2c and 2d, as well as Supplementary Table 4:

Revised Figures: Figures 2c and 2d have been updated to explicitly indicate which data points correspond to simple sorption processes and which represent sorption coupled with reduction. This revision aims to provide a clearer visual representation, aiding in the accurate interpretation of the mechanisms at play.

Fig. 2 Gold recovery performances. (a) The calculated Q_m and k_f of pyrocarbon from Langmuir model and mass transfer model. (b) adsorption kinetics of Au(III) using PyC700. (c) The comparison of Au(III) recovery capacity of PyC700 with that of previously published sorbents. (d) Effects of solution acid-alkaline conditions on Au(III) recovery by PyC700. (e) Effects of potentially interfering species including cations and anions on Au(III) recovery by PyC700. (f) The calculated K_d and the comparison of redox potentials for various metal ions. (g) Schematic of the gold recovery

from CPU leachate using PyC700. The concentration change of Au(III) using PyC700 as sorbent in (h) AMD leachate and (i) Intel leachate. (j) The Au(III) recovery by PyC700 in AMD and Intel leachate systems. The abbreviations “Sorp.” and “Sorp. & Red.” in Fig. 2c and 2d signify that the reported sorbents functionalized through simple sorption and sorption-coupled reduction processes, respectively.

Updated Supplementary Table 4: We have meticulously revised Supplementary Table 4 (See the table in the specific **Response to the Comment 1** of reviewer #2) to include additional columns or annotations that specify the nature of the process—whether it is pure sorption or sorption-reduction. Each entry now clearly states the underlying mechanism, ensuring that the basis for comparison is transparent and justifiable.

To clarify the performance comparison analysis, we also revised the relevant description in the manuscript as following:

Line 204: “This exceptional capacity surpasses other carbon-based sorbents (*e.g.*, activated carbon, carbon nanotube, graphene) and advanced nano sorbents (*e.g.*, MOFs, COFs, and metal sulfides), regardless of whether they rely on single sorption force or combined sorption-coupled reduction mechanisms (Supplementary Table 4).”

Overall, through these comprehensive revisions, we aim to provide a more nuanced and accurate portrayal of our findings, addressing the reviewer's concerns and enhancing the overall clarity and reliability of our comparative analysis.

References

1. Zhu, C.; Fang, Q.; Liu, R.; Dong, W.; Song, S.; Shen, Y., Insights into the Crucial Role of Electron and Spin Structures in Heteroatom-Doped Covalent Triazine Frameworks for Removing Organic Micropollutants. *Environ. Sci. Technol.* **2022**, *56*, (10), 6699-6709.
2. Xiang, C.; Ji, Q.; Zhang, G.; Wang, H.; Qu, J., In Situ Creation of Oxygen Vacancies in Porous Bimetallic La/Zr Sorbent for Aqueous Phosphate: Hierarchical Pores Control Mass Transport and Vacancy Sites Determine Interaction. *Environ. Sci. Technol.* **2020**, *54*, (1), 437-445.
3. Hu, W.; Yang, L.; Shao, P.; Shi, H.; Chang, Z.; Fang, D.; Wei, Y.; Feng, Y.; Huang, Y.; Yu, K.; Luo, X., Proton Self-Enhanced Hydroxyl-Enriched Cerium Oxide for Effective Arsenic Extraction from Strongly Acidic Wastewater. *Environ. Sci. Technol.* **2022**, *56*, (14), 10412-10422.
4. Chen, Y.; Qiao, Q.; Cao, J.; Li, H.; Bian, Z., Precious metal recovery. *Joule* **2021**, *5*, (12), 3097-3115.
5. Ilyas, S.; Anwar, M. A.; Niazi, S. B.; Afzal Ghauri, M., Bioleaching of metals from electronic scrap by moderately thermophilic acidophilic bacteria. *Hydrometallurgy* **2007**, *88*, (1), 180-188.
6. Lin, X.; Tran, D. T.; Song, M.-H.; Yun, Y.-S., Development of polyethyleneimine-starch fibers stable over the broad pH range for selective adsorption of gold from actual leachate solutions of waste electrical and electronic equipment. *J. Clean. Prod.* **2021**, *328*, 129545.
7. Shu, Y.; Chen, Y.; Han, Q.; Liu, X.; Liu, B.; Wang, Z., Selective and Light-Enhanced Au (III) Recovery by a Porphyrin-Based Metal–Organic Framework: Performance and Underlying Mechanisms. *ACS ES&T Engineering* **2023**, *3*, (7), 1042-1052.
8. Xue, T.; He, T.; Peng, L.; Syzgantseva, O. A.; Li, R.; Liu, C.; Sun, D. T.; Xu, G.; Qiu, R.; Wang, Y., A customized MOF-polymer composite for rapid gold extraction from water matrices. *Sci. Adv.* **2023**, *9*, (13), eadg4923.
9. Cao, J.; Xu, Z.; Chen, Y.; Li, S.; Jiang, Y.; Bai, L.; Yu, H.; Li, H.; Bian, Z., Tailoring the Asymmetric Structure of NH₂-UiO-66 Metal-Organic Frameworks for Light-promoted Selective and Efficient Gold Extraction and Separation. *Angew. Chem.* **2023**, *135*, (18), e202302202.
10. Sun, T.; Yen, W., Kinetics of gold chloride adsorption onto activated carbon. *Miner. Eng.* **1993**, *6*, (1), 17-29.
11. Simanova, S.; Shukarev, A.; Lysenko, A.; Grebennikov, S.; Astashkina, O., Adsorption of palladium, platinum, and gold chloride complexes by carbon fibers with various structures. *Fibre chemistry* **2008**, *40*, (4).
12. Behbahani, M.; Gorji, T.; Mahyari, M.; Salarian, M.; Bagheri, A.; Shaabani, A., Application of Polypropylene Amine Dendrimers (POPAM)-Grafted MWCNTs Hybrid Materials as a New Sorbent for Solid-Phase Extraction and Trace Determination of Gold(III) and Palladium(II) in Food and Environmental Samples. *Food Anal. Method.* **2013**, *7*, (5), 957-966.

13. Pang, S.-K.; Yung, K.-C., Prerequisites for achieving gold adsorption by multiwalled carbon nanotubes in gold recovery. *Chem. Eng. Sci.* **2014**, *107*, 58-65.
14. Wang, L.; Li, J.; Zhong, G.; Li, J.; Lu, X.; Wang, S.; Tang, Y., Diatom Biochar Recovered Au(III) Efficiently from Both Synthetic and Real Electroplating Wastewaters. *ACS ES&T Water* **2023**, *3*, (5), 1395-1405.
15. Chand, R.; Watari, T.; Inoue, K.; Kawakita, H.; Luitel, H. N.; Parajuli, D.; Torikai, T.; Yada, M., Selective adsorption of precious metals from hydrochloric acid solutions using porous carbon prepared from barley straw and rice husk. *Miner. Eng.* **2009**, *22*, (15), 1277-1282.
16. Liu, L.; Liu, S.; Zhang, Q.; Li, C.; Bao, C.; Liu, X.; Xiao, P., Adsorption of Au(III), Pd(II), and Pt(IV) from Aqueous Solution onto Graphene Oxide. *J. Chem. Eng. Data* **2012**, *58*, (2), 209-216.
17. Wang, Z.; Li, X.; Liang, H.; Ning, J.; Zhou, Z.; Li, G., Equilibrium, kinetics and mechanism of Au³⁺, Pd²⁺ and Ag⁺ ions adsorption from aqueous solutions by graphene oxide functionalized persimmon tannin. *Mater. Sci. Eng. C* **2017**, *79*, 227-236.
18. Liu, L.; Li, C.; Bao, C.; Jia, Q.; Xiao, P.; Liu, X.; Zhang, Q., Preparation and characterization of chitosan/graphene oxide composites for the adsorption of Au (III) and Pd (II). *Talanta* **2012**, *93*, 350-357.
19. Li, F.; Zhu, J.; Sun, P.; Zhang, M.; Li, Z.; Xu, D.; Gong, X.; Zou, X.; Geim, A.; Su, Y., Highly efficient and selective extraction of gold by reduced graphene oxide. *Nat. Commun.* **2022**, *13*, (1), 4472.
20. Hong, Y.; Thirion, D.; Subramanian, S.; Yoo, M.; Choi, H.; Kim, H. Y.; Stoddart, J. F.; Yavuz, C. T., Precious metal recovery from electronic waste by a porous porphyrin polymer. *Proc. Natl. Acad. Sci. U.S.A.* **2020**, *117*, (28), 16174-16180.
21. Ma, J.; Zou, Y.; Jiang, Z.; Huang, W.; Li, J.; Wu, G.; Huang, Y.; Xu, H., An in situ XAFS study—the formation mechanism of gold nanoparticles from X-ray-irradiated ionic liquid. *Phys. Chem. Chem. Phys.* **2013**, *15*, (28), 11904-11908.
22. Li, M.; Yao, Z.; Chen, Y.; Li, D.; Shao, J.; Dong, H.; Meng, Z.; Yang, L.; Ren, W.; Luo, X., Potential-dependent selectivity for the efficient capture of gold from E-waste acid leachate using sulfhydryl-functionalized carbon. *Sci. Bull.* **2023**, *68*, (11), 1095-1099.
23. Thanh, N. T. K.; Maclean, N.; Mahiddine, S., Mechanisms of nucleation and growth of nanoparticles in solution. *Chem. Rev.* **2014**, *114*, (15), 7610-7630.
24. Sun, D. T.; Gasilova, N.; Yang, S.; Oveisi, E.; Queen, W. L., Rapid, selective extraction of trace amounts of gold from complex water mixtures with a metal–organic framework (MOF)/polymer composite. *J. Am. Chem. Soc.* **2018**, *140*, (48), 16697-16703.
25. Feng, B.; Yao, C.; Chen, S.; Luo, R.; Liu, S.; Tong, S., Highly efficient and selective recovery of Au (III) from a complex system by molybdenum disulfide nanoflakes. *Chem. Eng. J.* **2018**, *350*, 692-702.

REVIEWER COMMENTS

Reviewer #1 (Remarks to the Author):

The authors were revised the manuscript based on the reviewers remarks to improve the manuscript. This can be accepted.

Reviewer #2 (Remarks to the Author):

The authors have placed significant effort into additional data and analysis onto the paper. However, the novelty of studying the adsorption of gold onto carbon surfaces is still unclear, as there are extensive industrially-relevant processes which use activated carbon for gold removal.

Furthermore, there is concern in the rebuttal when referencing gold aggregation as a "polymerization" (e.g. self-polymerization), which are often referred to units with organic moieties, rather than metal ions. While the stepwise nucleation terminology is acceptable, the polymerization characterization of the mechanism is most likely misused.

The efficient performance of the system and the additional analysis in the review have made this an excellent publication for a specialized technical journal in sustainability or waste removal, the novelty or impact of a pyrolysis-created carbon for gold removal is still not considered sufficient for Nature Communications.

Reviewer #3 (Remarks to the Author):

The detailed analysis and experimentation conducted in this paper demonstrate a deep understanding of the process of gold retrieval from e-waste leachate. Impressively, the authors have provided solid evidence to support their statement about the "stepwise" nucleation mechanism during the Au(III) reduction process, which has been neglected by previous work and can significantly enhance the understanding of Au(III) reduction theory. The paper offers clear and concise explanations of the methodology used, as well as the results obtained, which demonstrate a high level of gold recovery efficiency. The paper is well-structured, with excellent figures and a strong presentation of ideas. Overall, the research presented in this paper is well-executed and has the potential to have a significant impact on both the recycling industry and environmental conservation efforts. I recommend its acceptance with some minor concerns that I am given below:

1. Except for aqua regia, cyanide, the mixed solution of Cu-NH₃-S₂O₃-H₂O have been regarded as

alternative leaching system for gold in E-waste. And the $\text{Au}(\text{S}_2\text{O}_3)_2^{3-}$ will become the main existing form of the leached Au, so it was recommended to investigate the recovery capacity of the PyC700 for this specific type of gold ions.

2. Upon adsorbing and reducing the Au(III) on the sorbent, gold nanoparticles constantly form and agglomerate on the adsorption surfaces. How does this affect the electronic conductivity? This observation could provide valuable support for the concept of "autocatalysis growth".

3. The authors presented kinetics studies in Supplementary Fig. 9, with a gold concentration of approximately 43 ppm. However, it was found that within real leaching solutions from CPUs, the gold concentration can reach levels as high as 650 to 1330 ppm. In such conditions, how much time would be required for the sorbent to achieve sorption equilibrium? This information is crucial if the sorbent is used in mixed reactors or fixed bed equipment.

Reviewer #1

General comment: *The authors were revised the manuscript based on the reviewers remarks to improve the manuscript. This can be accepted.*

Response: We express our sincere gratitude to the reviewers for the meticulous review and profound insights. The invaluable comments provided have significantly enhanced the article's quality.

Reviewer #2

General comment: *The efficient performance of the system and the additional analysis in the review have made this an excellent publication for a specialized technical journal in sustainability or waste removal, the novelty or impact of a pyrolysis-created carbon for gold removal is still not considered sufficient for Nature Communications.*

Response: We extend our deepest gratitude to the reviewer for your detailed and thorough feedback, which plays an essential role in maintaining the high standards of articles published in *Nature Communications*. Please refer to our detailed responses regarding the specific comments in the subsequent sections.

Special comments:

1. The authors have placed significant effort into additional data and analysis onto the paper. However, the novelty of studying the adsorption of gold onto carbon surfaces is still unclear, as there are extensive industrially-relevant processes which use activated carbon for gold removal.

Response: In the *Previous Version of Response to Referees*, we have provided a comprehensive exposition on the novelty of our research, encompassing various innovative aspects of material designs and experimental methodologies, with detailed elucidation of the novel mechanisms for gold reduction and nucleation on PyC700. We fully acknowledge the reviewer's novelty concerns and are committed to conducting a thorough and rigorous examination to distinguish PyC700 sorbent from industrially used activated carbon in order to address your concerns.

The use of activated carbon for gold recovery has been widely acknowledged for its process and economic advantages, and it has been previously applied in real gold mining applications.¹ However, relevant research in this area has stagnated due to high difficulty in further improving the performance of activated carbons in gold recovery. Previous studies have shown that the recovery capacities of typical activated carbons were relatively low (<500 mg g⁻¹) despite their high specific surface area and reductive activity (Table R1). And a series of activated carbon-based Au(III) sorbents are summarized in Table R1 for performance comparison. It indicates that our PyC700 exhibits far superior gold recovery performance compared to other activated carbons in terms of gold capacity, kinetics, and sorption selectivity. In recent years, the development of nanotechnologies and material science has yielded a wide range of high-performance nano-sorbents that demonstrate promising potential for gold recovery, thereby diminishing interest in activated

carbon-based gold adsorption technology. However, the exorbitant preparation costs and intricate synthesis processes associated with these nano-materials have created an urgent demand for cost-effective yet efficient gold sorbents, such as activated carbons. Our work aims to identify and meticulously manipulate key structural/chemical properties of pyrocarbon sorbent to significantly improve its gold recovery performance. More importantly, we have demonstrated the significance of both carbon hybridization degree (sp^2 or sp^3) and surface hydroxyl groups, while systematically elucidating their crucial role through comprehension of the gold recovery mechanisms. This work revolutionizes traditional research focus on manipulating carbon properties solely by tuning pore structures or surface area. In comparison to other advanced nanomaterial sorbents (*e.g.*, molybdenic sulfide, metal-organic frameworks, covalent organic frameworks, and porphyrin polymers), PyC700 also demonstrates its highly profitable and readily separable merits make it particularly applicable for practical gold extraction from industrial E-waste leaching solution. The satisfactory technological advancements achieved in this study not only provide a feasible and cost-effective technique for gold recycling from waste, but also offer novel insights to rejuvenate traditional methodologies for addressing contemporary global challenges, including environmental concerns, resource recovery, and economic issues. Therefore, this study distinguishes itself significantly from conventional activated carbon gold adsorption in terms of conceptualization, methodology, and universality, aiming to foster advancements in material design, circular economy, and sustainability. We are confident that our work not only embodies significant innovation but also aligns seamlessly with the editorial focus of *Nature Communications*.

2. There is concern in the rebuttal when referencing gold aggregation as a "polymerization" (e.g. self-polymerization), which are often referred to units with organic moieties, rather than metal ions. While the stepwise nucleation terminology is acceptable, the polymerization characterization of the mechanism is most likely misused.

Response: We sincerely appreciate the reviewer's suggestion regarding the correction of terminology. The term "polymerization" has been employed in reference to previous studies,^{2,3} such as "chloride-coated Au-Cl polymers" and "the higher polymers and small Au clusters (such as Au₁₃ cluster)". Based on the available characterization results, we acknowledge that employing the term "stepwise nucleation" may be more appropriate for describing the gold reduction processes.

To avoid any misrepresentation, we have revised the relevant descriptions in the manuscript as shown below:

Line 42 to 44: “In details, the gold ions are reduced through stepwise nucleation processes of intermediate gold-chlorine clusters, which facilitates rapid gold reduction by lowering energy barriers from 1.08 to -21.84 eV.”;

Line 117 to 118: “This approach also revealed a novel “stepwise” reduction mechanism for gold nucleus formation.”;

Line 457 to 459: “Based on DFT calculations, the reduction of gold ions is proposed to occur through the stepwise nucleation of chlorine-protected gold clusters, which is crucial in gradually lowering the energy barriers for the entire reduction processes.”;

References

1. Buah, W. K.; Williams, P. T., Activated carbons prepared from refuse derived fuel and their gold adsorption characteristics. *Environ. Technol.* **2010**, *31*, (2), 125-137.
2. Ma, J.; Zou, Y.; Jiang, Z.; Huang, W.; Li, J.; Wu, G.; Huang, Y.; Xu, H., An in situ XAFS study—the formation mechanism of gold nanoparticles from X-ray-irradiated ionic liquid. *Phys. Chem. Chem. Phys.* **2013**, *15*, (28), 11904-11908.
3. Yao, T.; Sun, Z.; Li, Y.; Pan, Z.; Wei, H.; Xie, Y.; Nomura, M.; Niwa, Y.; Yan, W.; Wu, Z., Insights into initial kinetic nucleation of gold nanocrystals. *J. Am. Chem. Soc.* **2010**, *132*, (22), 7696-7701.

Table R1. Comparison of the PyC700 with previously reported activated carbon sorbent for Au(III) recovery.

Industrial activated carbon	Preparation or Resource	Conductivity (S m ⁻¹)	Surface area (m ² g ⁻¹)	Q _m (mg g ⁻¹)	K _d (mL g ⁻¹)	Equilibrium time (min)	Working pH	Mechanism	Ref.
HSAS	Activated carbon pyrolyzed from hard shell of apricot stones	0.3	1387	5.56	N.A.	90	N.A.	S	1
CECA C	Activated carbon from CECA company	0.05	1050	6.39	N.A.	N.A.	N.A.	S	1
Norit ROX 0.8	Activated carbon purchased from Brenntag Poland	N.A.	1570	N.A.	N.A.	30	N.A.	S&R	2
Activated carbon	Activated carbon produced from pyrolysis and steam gasification of refuse derived fuel	N.A.	500	32.1	N.A.	900	N.A.	S&R	3
CS700	Activated carbon pyrolyzed from corn straw	N.A.	N.A.	N.A.	N.A.	300	1.0~7.0	N.A.	4
H ₂ SO ₄ -AC	Activated carbon from pre-carbonization and H ₂ SO ₄ -activation of agro-waste biomass	N.A.	1211	479	N.A.	300	N.A.	S&R	5
PyC700	Pyrolyzed from calcium alginate followed by HCl rinsing	329	598	2829.7	3.1×10 ⁸	15	0.0~8.0	S&R	This work

[a] The term “S&R” suggests that a sorption-coupled reduction mechanism governs the entire gold recovery process rather than simple sorption.

[b] The term “S” indicates that the gold recovery process relies solely on sorption processes without any valence change for enriched gold ions.

References

1. Buah, W. K.; Williams, P. T., Activated carbons prepared from refuse derived fuel and their gold adsorption characteristics. *Environ. Technol.* **2010**, *31*, (2), 125-137.
2. Soleimani, M.; Kaghazchi, T., Adsorption of gold ions from industrial wastewater using activated carbon derived from hard shell of apricot stones—An agricultural waste. *Bioresour. Technol.* **2008**, *99*, (13), 5374-5383.
3. Wojnicki, M.; Luty-Błoch, M.; Socha, R. P.; Mech, K.; Pędzich, Z.; Fitzner, K.; Rudnik, E., Kinetic studies of sorption and reduction of gold (III) chloride complex ions on activated carbon Norit ROX 0.8. *Journal of Industrial and Engineering Chemistry* **2015**, *29*, 289-297.
4. Zhou, W.; Liang, H.; Lu, Y.; Xu, H.; Jiao, Y., Adsorption of gold from waste mobile phones by biochar and activated carbon in gold iodized solution. *Waste Manag.* **2021**, *120*, 530-537.
5. Bediako, J. K.; Kudoahor, E.; Lim, C.-R.; Affrifah, N. S.; Kim, S.; Song, M.-H.; Repo, E., Exploring the insights and benefits of biomass-derived sulfuric acid activated carbon for selective recovery of gold from simulated waste streams. *Waste Manag.* **2024**, *177*, 135-145.

Reviewer #3

General comment: The detailed analysis and experimentation conducted in this paper demonstrate a deep understanding of the process of gold retrieval from e-waste leachate. Impressively, the authors have provided solid evidence to support their statement about the "stepwise" nucleation mechanism during the Au(III) reduction process, which has been neglected by previous work and can significantly enhance the understanding of Au(III) reduction theory. The paper offers clear and concise explanations of the methodology used, as well as the results obtained, which demonstrate a high level of gold recovery efficiency. The paper is well-structured, with excellent figures and a strong presentation of ideas. Overall, the research presented in this paper is well-executed and has the potential to have a significant impact on both the recycling industry and environmental conservation efforts. I recommend its acceptance with some minor concerns that I am given below:

Response: We express our utmost gratitude to the reviewer for their highly positive feedback and comprehensive insights. The concerns raised by the reviewer have been duly addressed, and we have provided meticulous point-by-point responses. Please refer to our detailed responses regarding the specific comments in the subsequent sections.

Special comments:

1. Except for aqua regia, cyanide, the mixed solution of Cu-NH₃-S₂O₃-H₂O have been regarded as alternative leaching system for gold in E-waste. And the Au(S₂O₃)₂³⁻ will become the main existing form of the leached Au, so it was recommended to investigate the recovery capacity of the PyC700 for this specific type of gold ions.

Response: Thanks for the comment. Following this comment, we have performed additional experiments to evaluate the recovery capacity of Au(I) as Au(S₂O₃)₂³⁻ species on the PyC700. The relevant results were further provided in Supplementary Fig. 17a and Table 6. It indicates that the maximum sorption capacity (Q_m) of Au(I) as Au(S₂O₃)₂³⁻ species was calculated to be 252.6 mg g⁻¹ from the Langmuir model fitting of isotherm data. This Q_m value for Au(S₂O₃)₂³⁻ is relatively low with a drastic decrease by 91.1% compared with that of AuCl₄⁻. The OPCT curve on PyC700 electrode in the Au(S₂O₃)₂³⁻ solution was also measured, suggesting negligible electron transfer in these adsorption systems (Supplementary Fig. 17b). Unlike AuBr₄⁻ and AuCl₄⁻, both Au(CN)₂⁻ and Au(S₂O₃)₂³⁻ have extremely low reduction potentials, rendering high difficulty to be reduced by the PyC700, as manifested by the XRD in Supplementary Fig. 17d. These results further rendered the concept of "Reduction potential-dependent recovery behaviors" proposed in our work (Supplementary Fig. 17c).

Supplementary Fig. 17. (a) Adsorption isotherms for Au(CN)₂⁻, Au(S₂O₃)₂³⁻, and AuBr₄⁻ adsorption onto PyC700, (b) OCPT curves on the PyC700 electrodes in various Au(III) species solution systems. (c) maximum recovery capacity for various gold complexes with different reduction potential (E_0). (d) XRD patterns of the PyC700 after the gold sorption in 500 mg L⁻¹ AuCl₄⁻, AuBr₄⁻, Au(S₂O₃)₂³⁻, and Au(CN)₂⁻ solution, respectively.

Supplementary Table 6. Isotherm fitting of Langmuir model for different gold ions adsorption by PyC700.

Gold ions species	Temperature (°C)	Langmuir model		
		q_m (mg g ⁻¹)	k_L (L mg ⁻¹)	R ²
AuCl ₄ ⁻	25	2829.7	5.72	0.9184
AuBr ₄ ⁻	25	934.9	0.22	0.9849
Au(S ₂ O ₃) ₂ ³⁻	25	252.6	0.02	0.9426
Au(CN) ₂ ⁻	25	121.3	0.02	0.9326

To reflect these results, Supplementary Fig. 17 and Table 6 has been revised, and we have also revised the relevant discussion in the main text as follows:

Line 245 to 259: “Considering various leaching agents commonly employed in hydrometallurgy, we conducted additional investigations to assess the impacts of prevalent forms of gold complexes,⁵⁸ such as AuBr_4^- and $\text{Au}(\text{CN})_2^-$, and $\text{Au}(\text{S}_2\text{O}_3)_2^{3-}$. Our findings revealed a moderate decrease in measured Q_m values by 66.9% for AuBr_4^- , while significant decreases of 91.1% and 95.7% was observed for $\text{Au}(\text{S}_2\text{O}_3)_2^{3-}$ and $\text{Au}(\text{CN})_2^-$, respectively (Supplementary Fig. 17a and Table 6). The hypothesis that adsorption behaviors depend on reduction potential effectively clarifies this phenomenon (Supplementary Fig. 17b, c). It suggests that transferring electrons to AuBr_4^- becomes increasingly difficult, and it's even less probable for $\text{Au}(\text{S}_2\text{O}_3)_2^{3-}$ and $\text{Au}(\text{CN})_2^-$ to be reduced to metallic Au^0 (Supplementary Fig. 17d). This phenomenon can also be attributed to the differences in the coordination stability constants ($\lg K_f^0$) among these three complexes with Au-anions complexes ($\lg K_f^0(\text{Au}(\text{CN})_2^-) > \lg K_f^0(\text{Au}(\text{S}_2\text{O}_3)_2^{3-}) > \lg K_f^0(\text{AuBr}_4^-) > \lg K_f^0(\text{AuCl}_4^-)$), as higher $\lg K_f^0$ value indicates greater difficulty in decomposing the ligands from Au-anions to form metallic Au-Au bonds.⁵⁹”

2. Upon adsorbing and reducing the Au(III) on the sorbent, gold nanoparticles constantly form and agglomerate on the adsorption surfaces. How does this affect the electronic conductivity? This observation could provide valuable support for the concept of "autocatalysis growth".

Response: We appreciated the reviewer for the comment. To examine the effect of gold agglomeration on the conductivity (σ) of PyC700, we have prepared a series of PyC700 samples by adsorbing 500 mg L^{-1} Au(III) at varying reaction times. Their σ values were measured using four-point probe method with a resistivity tester. As shown in Supplementary Fig. 30, the σ values of the PyC700 substantially increased from 566 to 856 S m^{-1} during the first 30 min of adsorption, then reached a plateau after reaction for 120 min with σ values over than 2400 S m^{-1} . It is conceivable that the increased conductivity of the sorbent originates from the consecutive recovered element gold nanoparticles aggregated on the sorbent surfaces, which can further promote the electrons transfer at the adsorption interfaces to facilitate the gold recovery. Additionally, those reduced Au seeds exhibit a high surface free energy, rendering them unstable in a 'boiling state' that promotes the consumption of Au(III) salt and facilitates electron transfer for the aggregation of Au^0 NPs. Therefore, it is hypothesized that both enhanced reactivity and electron transfer contribute to efficient Au(III) recovery on the PyC700 sorbent, which can be referred to as autocatalytic surface growth according to previous studies.^{1,2}

To reflect this crucial aspect, we have provided the conductivity results as Supplementary Fig. 30:

Supplementary Fig. 30. The electronic conductivity of the PyC700 sample that adsorbed Au(III) for varying reaction times. After reaction in $500\ mg\ L^{-1}$ $AuCl_4^-$ solution for certain period, the PyC700 samples were freezing-dried, and their conductivities were measured via four-point probe method with resistivity tester (Malvern Mastersizer 2000). Based on the XRD and SEM results (Supplementary Fig. 10 and 29), we observed a consecutive reduction of Au(III) salt to elemental gold during the adsorption processes, followed by the formation of nanoscale Au seeds. These Au seeds exhibit a high surface free energy, rendering them unstable in a 'boiling state' that promotes the consumption of Au(III) salt and facilitates electron transfer for the aggregation of Au^0 NPs. Our findings in Supplementary Fig. 30 further demonstrate a significant enhancement in PyC700 conductivity due to the aggregation of Au^0 NPs on its surfaces during the initial gold recovery stage. Therefore, it is hypothesized that both enhanced reactivity and electron transfer contribute to efficient Au(III) recovery on the PyC700 sorbent, which can be referred to as autocatalytic surface growth based on previous studies.

The description in the main text of manuscript has also been revised as follows:

Line 376 to 380: “Notably, these gold seeds exhibit a significantly high surface free energy,^{34, 64} rendering them inherently unstable in a state of boiling that facilitates the consumption of Au(III) salt. Simultaneously, the enhanced PyC700 conductivity further promotes efficient electron transfer for subsequent reduction of gold ions (Supplementary Fig. 30).”.

3. The authors presented kinetics studies in Supplementary Fig. 9, with a gold concentration of approximately 43 ppm. However, it was found that within real leaching solutions from CPUs, the gold concentration can reach levels as high as 650 to 1330 ppm. In such conditions, how much time would be required for the sorbent to achieve sorption equilibrium? This information is crucial if the sorbent is used in mixed reactors or fixed bed equipment.

Response: To address the reviewer's concern, we expanded our kinetics studies to include scenarios in real CPUs leaching solutions. Notably, the AMD-Aqua leachate without dilution was excluded for a low recovery efficiency < 42.5% due to its highly acidic nature (pH < 0.0) and strong oxidation characteristics, as previously discussed in the main text. Gold concentrations range from ~2.7 to 650 mg L⁻¹ in four objective leaching solutions (*e.g.*, AMD-Aqua*10, AMD-NBS, Intel-Aqua, and Intel-NBS). As illustrated in Supplementary Fig. 18, PyC700 exhibits rapid gold adsorption kinetics at both high and relatively low concentration ranges. It reaches adsorption equilibrium within ~180 and 600 min at high concentration of 133 and 650 mg L⁻¹ in AMD-Aqua*10 and AMD-NBS, respectively. Meanwhile, over than 98.0% of Au(III) can be successfully recovered from Intel-Aqua and Intel-NBS leachates within 30 min. These kinetics results verified rapid adsorption capability of the PyC700 in real E-waste leachates, showcasing its high prospect in practical adsorption equipment (*e.g.*, mixed reactors or fixed bed column).

Supplementary Fig. 18. Kinetics curves for the Au(III) adsorption on PyC700 in four real CPUs leaching solutions of (a) AMD-Aqua*10, (b) AMD-NBS, (c) Intel-Aqua, and (d) Intel-NBS.

PyC700 exhibits rapid adsorption kinetics at both high and relatively low concentration ranges. It reaches adsorption equilibrium within ~180 and 600 min at high Au(III) concentration of 133 and 650 mg L⁻¹ in AMD-Aqua*10 and AMD-NBS leachates, respectively. Meanwhile, over than 98.0% of Au(III) can be successfully recovered from Intel-Aqua and Intel-NBS leachates within 30 min. The kinetics results verified rapid adsorption capability of the PyC700 in real E-waste leachates, showcasing its high prospect in practical adsorption equipment (*e.g.*, mixed reactors or fixed bed column).

To reflect these results, we have supplemented the kinetics data as Supplementary Fig. 18.

We also revised the relevant description in the main text of manuscript as follows:

Line 271 to 275: “As a result, PyC700 reached an adsorption equilibrium less than 300 min and successfully recovered nearly 100.0% of Au(III) in AMD-Aqua*10 leachate (Supplementary Fig. 18), reducing the residual concentration to about 0.1 µg L⁻¹, demonstrating profound recovery capabilities in practical applications.”

References

1. Yang, T.-H.; Zhou, S.; Gilroy, K. D.; Figueroa-Cosme, L.; Lee, Y.-H.; Wu, J.-M.; Xia, Y., Autocatalytic surface reduction and its role in controlling seed-mediated growth of colloidal metal nanocrystals. *Proc. Natl. Acad. Sci. U.S.A.* **2017**, *114*, (52), 13619-13624.
2. Thanh, N. T. K.; Maclean, N.; Mahiddine, S., Mechanisms of nucleation and growth of nanoparticles in solution. *Chem. Rev.* **2014**, *114*, (15), 7610-7630.

REVIEWERS' COMMENTS

Reviewer #3 (Remarks to the Author):

The manuscript has been improved greatly and the authors give positive responses to the comments. It can be accepted.